# Conformal prediction under ambiguous ground truth

**David Stutz**[1]**, Abhijit Guha Roy**[2]**, Tatiana Matejovicova**[1]**, Patricia Strachan**[2]**, Ali Taylan Cemgil**[1]**, Arnaud Doucet**[1]

[1]Google DeepMind, [2]Google
{dstutz, agroy, tatianama, trishs, taylancemgil, arnauddoucet}@google.com

**Reviewed on OpenReview:** https://openreview.net/forum?id=CAd6V2qXxc

## Abstract

Conformal Prediction (CP) allows to perform rigorous uncertainty quantification by constructing a prediction set $C(X)$ satisfying $\mathbb{P}(Y \in C(X)) \geq 1 - \alpha$ for a user-chosen $\alpha \in [0,1]$ by relying on calibration data $(X_1, Y_1), ..., (X_n, Y_n)$ from $\mathbb{P} = \mathbb{P}^X \otimes \mathbb{P}^{Y|X}$. It is typically implicitly assumed that $\mathbb{P}^{Y|X}$ is the "true" posterior label distribution. However, in many real-world scenarios, the labels $Y_1, ..., Y_n$ are obtained by aggregating expert opinions using a voting procedure, resulting in a one-hot distribution $\mathbb{P}^{Y|X}_{\text{vote}}$. This is the case for most datasets, even well-known ones like ImageNet. For such "voted" labels, CP guarantees are thus w.r.t. $\mathbb{P}_{\text{vote}} = \mathbb{P}^X \otimes \mathbb{P}^{Y|X}_{\text{vote}}$ rather than the true distribution $\mathbb{P}$. In cases with unambiguous ground truth labels, the distinction between $\mathbb{P}_{\text{vote}}$ and $\mathbb{P}$ is irrelevant. However, when experts do not agree because of ambiguous labels, approximating $\mathbb{P}^{Y|X}$ with a one-hot distribution $\mathbb{P}^{Y|X}_{\text{vote}}$ ignores this uncertainty. In this paper, we propose to leverage expert opinions to approximate $\mathbb{P}^{Y|X}$ using a non-degenerate distribution $\mathbb{P}^{Y|X}_{\text{agg}}$. We then develop *Monte Carlo CP* procedures which provide guarantees w.r.t. $\mathbb{P}_{\text{agg}} = \mathbb{P}^X \otimes \mathbb{P}^{Y|X}_{\text{agg}}$ by sampling multiple synthetic pseudo-labels from $\mathbb{P}^{Y|X}_{\text{agg}}$ for each calibration example $X_1, ..., X_n$. In a case study of skin condition classification with significant disagreement among expert annotators, we show that applying CP w.r.t. $\mathbb{P}_{\text{vote}}$ under-covers expert annotations: calibrated for 72% coverage, it falls short by on average 10%; our Monte Carlo CP closes this gap both empirically and theoretically. We also extend Monte Carlo CP to multi-label classification and CP with calibration examples enriched through data augmentation.

## 1 Introduction

Many application domains, especially safety-critical applications such as medical diagnostics, require reasonable uncertainty estimates for decision making and benefit from statistical performance guarantees. Conformal prediction (CP) is a statistical framework allowing to quantify uncertainty rigorously by providing finite-sample, non-asymptotic performance guarantees. First introduced by Vovk et al. (2005), it has become very popular in machine learning as it is widely applicable while making no dsitributional or model assumptions, e.g., see (Romano et al., 2019; Sadinle et al., 2019; Romano et al., 2020; Angelopoulos et al., 2021; Stutz et al., 2021; Fisch et al., 2022) and Angelopoulos & Bates (2021) for more references.

Specifically, we consider a classification task with $K$ classes and denote by $[K]$ the set $\{1, ..., K\}$. Then, a classifier $\pi : \mathcal{X} \to \Delta^K$ outputs the class probabilities where $\Delta^K$ is the $K$-simplex. Based only on a held-out set of $n$ calibration examples $(X_i, Y_i) \sim \mathbb{P}$, CP allows us to return a prediction set $C(X) \subseteq [K]$ for a given test point $(X, Y)$ (with $Y$ being unobserved) dependent on the calibration data such that

$$\mathbb{P}(Y \in C(X)) \geq 1 - \alpha, \tag{1}$$

whatever being $\mathbb{P}$ and $\pi$ as long as the joint distribution of $((X_1, Y_1), ..., (X_n, Y_n), (X, Y))$ is exchangeable. Here $\alpha \in [0, 1]$ is a user-specified parameter and the probability in Equation (1) is not only over $(X, Y)$ but

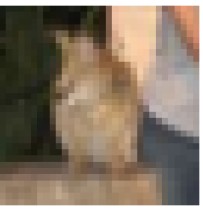 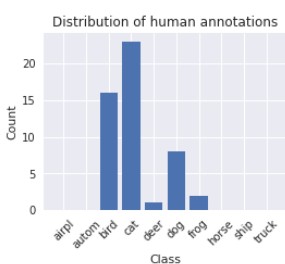 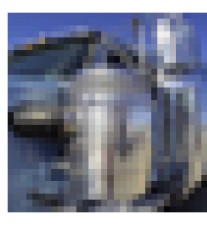 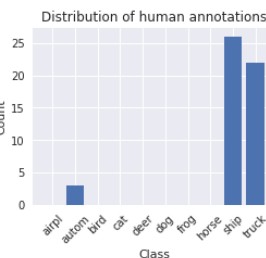

Figure 1: Two ambiguous examples from CIFAR10-H (Peterson et al., 2019) with the corresponding annotations of $p = 50$ experts. These annotations can be used to build $\mathbb{P}_{\text{agg}}(Y = y | X = x, Y^1, ..., Y^p)$, see text for details.

also over the calibration set. This is called the *coverage guarantee* of CP. The size of such prediction sets $|C(X)|$, also called *inefficiency*, is a good indicator of the uncertainty for $X$.

It is commonly taken for granted that $\mathbb{P} = \mathbb{P}^X \otimes \mathbb{P}^{Y|X}$ where $\mathbb{P}^{Y|X}$ is the "true" posterior label distribution so that the l.h.s. of Equation (1) is the probability for the ground truth label $Y$ being an element of the prediction set $C(X)$. However, in many practical applications, the calibration labels $(Y_i)_{i \in [n]}$ are obtained based on (multiple) expert opinions. In medical applications, for example, it is usually impossible to identify the patient's actual condition as this would require invasive and expensive tests. Instead, labels are derived from expert annotations, e.g., doctor ratings (Liu et al., 2020). Beyond medical diagnosis, however, this is generally the case for many popular datasets such as CIFAR10 (Krizhevsky, 2009; Peterson et al., 2019), COCO (Lin et al., 2014), ImageNet (Russakovsky et al., 2015) or many datasets from the GLUE benchmark (Wang et al., 2019) such as MultiNLI (Williams et al., 2018) (see Appendix A). Based on the expert annotations, the majority voted label is then typically selected as the ground truth label. Formally, this means that we consider $Y_i \sim \mathbb{P}_{\text{vote}}(\cdot | X = x_i)$ where $\mathbb{P}_{\text{vote}}(Y = y | X = x)$ is a one-hot distribution. Thus, when using CP on these *voted labels*, we obtain guarantees of the form $\mathbb{P}_{\text{vote}}(Y \in C(X)) \geq 1 - \alpha$ for $\mathbb{P}_{\text{vote}} = \mathbb{P}^X \otimes \mathbb{P}_{\text{vote}}^{Y|X}$ rather than a guarantee w.r.t. to the true distribution, $\mathbb{P}(Y \in C(X)) \geq 1 - \alpha$. The difference between $\mathbb{P}_{\text{vote}}$ and $\mathbb{P}$ is small for some tasks, including popular benchmarks such as CIFAR10, because most examples are fairly unambiguous, meaning that annotators do rarely disagree because $\mathbb{P}(Y = y | X = x)$ is one-hot anyway and a simple aggregation strategy can unambiguously determine the true class. However, in complex tasks such as the dermatology problem addressed in this work, there are many examples where experts disagree. More importantly, this disagreement is often "irresolvable" (Schaekermann et al., 2016; Gordon et al., 2022; Uma et al., 2022), meaning that obtaining more annotations will likely not reduce disagreement. In such cases, $\mathbb{P}_{\text{vote}}(Y = y | X = x)$ differs significantly from the true conditional distribution $\mathbb{P}(Y = y | X = x)$.

In such scenarios, performing CP using voted labels can severely underestimate the true uncertainty (see Section 2.2 for an intuitive example). Instead of summarizing the expert annotations by a single one-hot distribution, we propose here to rely on an approximation of $\mathbb{P}_{\text{agg}}(Y = y | X = x)$ to $\mathbb{P}(Y = y | X = x)$ to better account for this uncertainty. As a simple example, assume that for each calibration data $X$, expert $q \in [p]$ annotates a single class $Y^q \in [K]$ as done on CIFAR10-H (Peterson et al., 2019), see Figure 1 for an illustration. Then, let $p_k = \sum_{q=1}^{p} \mathbb{I}(Y^q = k)$ be the number of experts selecting class $k$, a simple voting procedure sets $\mathbb{P}_{\text{vote}}(Y = y | X = x, Y^1, ..., Y^p) = \mathbb{I}(y = \hat{Y})$ where $\hat{Y} = \arg\max_{k \in [K]} p_k$ (disregarding ties for simplicity) and unconditionally we also typically will have $\mathbb{P}_{\text{vote}}(Y = y | X = x)$ a one-hot distribution in unambiguous cases (i.e. the same label is selected for all expert annotations). Obviously, if there is sufficient disagreement among annotators, taking $\hat{Y}$ will ignore many of the (usually expensive) annotations. As a result, performing CP on the voted labels ignores this uncertainty. Instead, we argue that selecting an aggregated distribution, e.g., $\mathbb{P}_{\text{agg}}(Y = y | X = x, Y^1, ..., Y^p) = \frac{1}{p} \sum_{k \in [K]} p_k \mathbb{I}(y = k)$, allows us to better capture uncertainty present in the annotations. Ideally, by integrating over $(Y^1, ..., Y^p)$, the resulting $\mathbb{P}_{\text{agg}}$ is a good approximation of the true distribution $\mathbb{P}$ (and we discuss common ways of constructing $\mathbb{P}_{\text{agg}}$ in Section 3.1).

**Contributions:** In this paper, we propose CP procedures that allow us to construct a prediction set $C(X)$ satisfying $\mathbb{P}_{\text{agg}}(Y \in C(X)) \geq 1 - \alpha$ for $\mathbb{P}_{\text{agg}} = \mathbb{P}^X \otimes \mathbb{P}_{\text{agg}}^{Y|X}$[1] as long as one can sample from $\mathbb{P}_{\text{agg}}^{Y|X}$. Note that while it would be desirable to have instead guarantees w.r.t. the distribution $\mathbb{P}$, this is an impossible task as we never observe any data with labels sampled from $\mathbb{P}^{Y|X}$. Whether $\mathbb{P}_{\text{agg}}^{Y|X}$ is a good approximation to $\mathbb{P}^{Y|X}$ will be application dependent. Instead, the prediction set $C(X)$ outputted by our procedures is the one we could compute if the experts had access to $X$ and their opinions were aggregated through $\mathbb{P}_{\text{agg}}^{Y|X}$. This is the best one can hope for. We emphasize that we are not making more model assumptions than is currently made by applying CP to voted labels from $\mathbb{P}_{\text{vote}}^{Y|X}$. In contrast, we make the usually implicit assumptions on label collection explicit. To perform calibration with $\mathbb{P}_{\text{agg}}^{Y|X}$, we propose a sampling-based approach, coined **Monte Carlo CP**, which proceeds by sampling multiple "pseudo ground truth" labels from $\mathbb{P}_{\text{agg}}^{Y|X}$ for each calibration example $X_1, \ldots, X_n$. We show how this approach can provide rigorous coverage guarantees despite not having access to exchangeable calibration examples. We present experiments on a skin condition classification case study: Here, calibration with voted labels from $\mathbb{P}_{\text{vote}}^{Y|X}$ is shown to undercover expert annotated labels by a staggering 10%. Our approach closes this gap both theoretically and empirically. Moreover, we discuss extensions to multi-label classification and calibration with augmented calibration examples for robust CP.

**Outline:** The rest of this paper is structured as follows: In Section 2, we illustrate the problem on a toy dataset and introduce required background on CP. Then, Section 3 introduces our original Monte Carlo CP procedures, highlights the applicability of this approach to related problems and discusses related work. In Section 4, we present an application to skin condition classification following (Liu et al., 2020), multi-label classification and data augmentation before concluding in Section 5.

## 2 Background and Motivating Example

### 2.1 Conformal prediction

In the following, we briefly review standard (split) CP (Vovk et al., 2005; Papadopoulos et al., 2002). To this end, we assume a classifier $\pi_y(x) \approx \mathbb{P}(Y = y | X = x)$ approximating the posterior label probabilities is available. This model will be typically based on learned parameters using a training set. Then, given a set of calibration examples $(X_i, Y_i)_{i \in [n]}$ from $\mathbb{P}$, we want to construct a prediction set $C(X) \subseteq [K]$ for the test point $(X, Y)$ such that the coverage guarantee from Equation (1) holds. As mentioned earlier, this requires the calibration examples and the test example to be exchangeable but does not make any further assumption on the data distribution or on the underlying model $\pi$.

A popular conformal predictor proceeds as follows: given a real-valued conformity score $E(X, k)$ based on the model predictions $\pi(x) \in \Delta^K$, we define

$$C(X) = \{k \in [K] : E(X, k) \geq \tau\} \tag{2}$$

where $\tau$ is the $\lfloor \alpha(n+1) \rfloor$ smallest element of $\{E(X_i, Y_i)\}_{i \in [n]}$, equivalently $\tau$ is obtained by computing the $\lfloor \alpha(n+1) \rfloor / n$ quantile of the distribution of the conformity scores of the calibration examples

$$\tau = Q\left(\{E(X_i, Y_i)\}_{i \in [n]}; \frac{\lfloor \alpha(n+1) \rfloor}{n}\right). \tag{3}$$

Here, $Q(\cdot; q)$ denotes the $q$-quantile. By construction of the quantile, see e.g. (Romano et al., 2019; Vovk et al., 2005; Angelopoulos & Bates, 2021), this ensures that the lower bound on coverage in Equation (1) is satisfied. Additionally, if the conformity scores are almost surely distinct, then we have the following upper bound $\mathbb{P}(Y \in C(X)) \leq 1 - \alpha + \frac{1}{n+1}$. The conformity score is a design choice. A standard choice, which we will use throughout the paper, is $E(x, k) = \pi_k(x)$ (Sadinle et al., 2019) but many alternative scores have been proposed in the literature (Romano et al., 2020; Angelopoulos et al., 2021).

---

[1] As explained further, we cannot obtain i.i.d. samples from $\mathbb{P}_{\text{agg}}^{Y|X}$ but only exchangeable ones. This prevents the use of concentration inequalities to obtain bounds on $\mathbb{P}_{\text{agg}}(Y \in C(X) | X = x)$ for a given $C(X)$.

Figure 2: Illustration of CP on a toy dataset detailed in Section 2.2 for two examples $X$, indexed 0 and 1. Here, the conformity scores $\{E(X,k)\}_{k\in[K]}$ are taken to be the estimated posterior probabilities $\pi_k(X)$ of a multilayer preceptron (MLP) whose decision boundaries are shown. To construct the prediction sets $C(X)$, these scores are thresholded using the threshold $\tau$, cf. Equations (2) and (3). Equivalently, we can use the $p$-value associated formulation and threshold $(\rho_y)_{y\in[K]}$ at confidence level $\alpha$, cf Equations (4) and (5). Calibrating against true labels, both approaches obtain coverage $1-\alpha$ (w.r.t. the true labels).

An alternative view on CP can be obtained through a $p$-value formulation; see e.g. (Sadinle et al., 2019). The conformity scores of the calibration examples and test example $(X,Y)$ are exchangeable so if the distribution of $E(X,Y)$ is continuous then

$$\rho_Y = \frac{|\{i \in [n] : E(X_i, Y_i) \leq E(X,Y)\}| + 1}{n+1} = \frac{\sum_{i=1}^{n} \mathbb{I}[E(X_i, Y_i) \leq E(X,Y)] + 1}{n+1} \tag{4}$$

is uniformly distributed over $\{1/(n+1), 2/(n+1)..., 1\}$ and thus $\rho_Y$ is a $p$-value in the sense it satisfies $\mathbb{P}(\rho_Y \leq \alpha) \leq \alpha$, equivalently $\mathbb{P}(\rho_Y > \alpha) \geq 1-\alpha$. If the distribution is not continuous, then it can be checked that $\mathbb{P}(\rho_Y \leq \alpha) \leq \alpha$ still holds (see e.g. (Bates et al., 2023)). It follows directly that

$$C(X) := \{y \in [K] : \rho_y > \alpha\} \tag{5}$$

satisfies $\mathbb{P}(Y \in C(X)) \geq 1-\alpha$ and the prediction set obtained this way is identical to the one obtained using Equations (2) and (3). For completeness, the equivalence between both formulations is detailed in Section B.1 of Appendix B. In contrast to calibrating the threshold $\tau$, the p-value formulation requires computing $(\rho_k)_{k\in[K]}$ for each test example $X$ and is thus computationally more expensive. The $p$-value formulation will be useful in our context as $p$-values can easily be combined to obtain a new $p$-value; see e.g. (Vovk & Wang, 2020). We illustrate an application of CP to a 3-class classification problem in Figure 2.

## 2.2 Motivating Example

We now consider a toy dataset to illustrate the impact a voting strategy can have on prediction sets produced by CP. Consider the true distribution $\mathbb{P} = \mathbb{P}^X \otimes \mathbb{P}^{Y|X} = \mathbb{P}^Y \otimes \mathbb{P}^{X|Y}$ be defined as follows: $\mathbb{P}^{X|Y}$ admits a density $p(x|y) = \mathcal{N}(x; \mu_y, \mathrm{diag}(\sigma_y^2))$ with $\mu_y \in \mathbb{R}^d$ and $\mathrm{diag}(\sigma_y^2) \in \mathbb{R}^{d\times d}$ being mean and variance. Further, one has $\mathbb{P}(Y = y) = w_y$ with $\sum_{y=1}^{K} w_y = 1$. To generate examples $(X,Y) \sim \mathbb{P}$, we first sample $Y$ from $\mathbb{P}(Y = y) = w_y$ and then $X$ from $p(x|y)$ so we have ground truth labels for each example. Furthermore, by Bayes' rule we have access to the true posterior probability mass function $\mathbb{P}(Y = y|X = x)$. If the Gaussians are well-separated, $\mathbb{P}(Y = y|X = x)$ will be crisp, i.e., close to one-hot with low entropy. However, encouraging significant overlap between these Gaussians, e.g., by moving the means $(\mu_k)_{k\in[K]}$ close together, will result in highly ambiguous $\mathbb{P}(Y = y|X = x)$. We sample examples $(X_i, Y_i)_{i\in[n]}$ as outlined above and indicate classes by color in Figure 3 (top left). These synthetic data were previously used in the example displayed in Figure 2 to illustrate CP. The true posterior label probabilities $\mathbb{P}(Y_i = y|X = X_i)$ are also displayed, cf. Figure 3 (top middle). As can be seen, these distributions can be very ambiguous in between all three classes. Let us assume a large set of annotators that allow us by majority vote to recover $\hat{Y}_i = \arg\max_{y\in[K]} \mathbb{P}(Y = y|X = X_i)$, i.e., the voted label, as shown on Figure 3 (top right). This defines the one-hot distribution $\mathbb{P}_{\mathrm{vote}}(Y = y|X = X_i) = \mathbb{I}(y = \hat{Y}_i)$. Clearly, these voted labels ignore the fact that $\mathbb{P}^{Y|X}$ can have high entropy.

Contrary to Figure 2, we now perform standard split CP using calibration data relying on the voted labels $(X_i, \hat{Y}_i)_{i\in[n]}$ from $\mathbb{P}_{\mathrm{vote}} = \mathbb{P}^X \otimes \mathbb{P}_{\mathrm{vote}}^{Y|X}$ rather than $(X_i, Y_i)_{i\in[n]}$ from $\mathbb{P} = \mathbb{P}^X \otimes \mathbb{P}^{Y|X}$, using a conformity

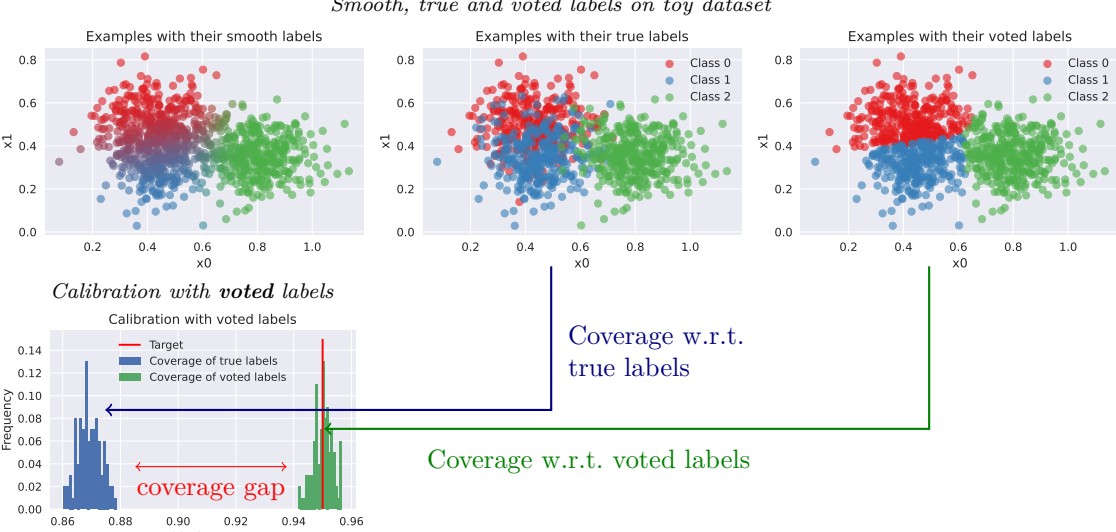

Figure 3: Illustration of an ambiguous problem with $K = 3$ classes, in two dimensions, using our toy dataset. Top: We show examples colored by their true posteriors $\mathbb{P}(Y = y | X = x)$ (left), true labels (middle) and voted labels, i.e., $\hat{Y} = \arg\max_{y \in [K]} \mathbb{P}(Y = y | X = x)$ (right). Note the high ambiguity between the classes, which is clearly ignored in the voted labels. Bottom: Empirical coverage over random calibration/test splits, i.e. the proportion of true or voted labels included in the constructed prediction sets $C(x)$, when calibrating against the voted labels. This produces prediction sets that significantly undercover w.r.t. true labels. Figure 4 shows how our Monte Carlo conformal prediction, as introduced in Section 3, overcomes this gap.

score $E(x, k) = \pi_k(x)$ where $\pi_k(x)$ is given by a multilayer perceptron. We randomly split the examples in two halves for calibration and testing. In Figure 3 (bottom), we plot the empirical coverage, i.e., the fraction of test examples for which (a) the true label (blue) or (b) the voted label (green) is included in the predicted prediction set. In this case, CP guarantees the latter by design, (b), to be 95% (on average across calibration/test splits). Strikingly, however, coverage against the (usually unknown) true labels is significantly worse. Of course, this gap depends on the ambiguity of the problem.

# 3 Monte Carlo conformal prediction

As discussed in the introduction and illustrated in Section 2.2, we want to avoid using a one-hot distribution $\mathbb{P}_{\text{vote}}^{Y|X}$ when labeling ambiguous examples. Instead we first explain here how, based on expert opinions, we can obtain non-degenerate estimates $\mathbb{P}_{\text{agg}}^{Y|X}$ of $\mathbb{P}^{Y|X}$. We then show how $\mathbb{P}_{\text{agg}}^{Y|X}$ can be leveraged to provide novel CP procedures for constructing prediction sets that satisfy coverage guarantees w.r.t. $\mathbb{P}_{\text{agg}} = \mathbb{P}^X \otimes \mathbb{P}_{\text{agg}}^{Y|X}$. Given that we never observe labels from the true distribution $\mathbb{P}^{Y|X}$, providing guarantees w.r.t. $\mathbb{P}_{\text{agg}}$ is the best we can hope to do. For a test example $X$, this can be understood as guaranteeing coverage against labels that expert annotators would assign if they had access to $X$ and their annotations were aggregating using $\mathbb{P}_{\text{agg}}^{Y|X}$.

## 3.1 From expert annotations to $\mathbb{P}_{\text{agg}}^{Y|X}$

From now on, we assume that we have calibration data $(X_i, B_i) \sim \mathbb{P}$ for $i \in [n]$ and a test data $X, B \sim \mathbb{P}$ with $B$ unobserved; the joint distribution of $(X_i, B_i)_{i \in [n]}$ and $(X, B)$ being exchangeable. Here $B_i$ corresponds to a set of expert annotations, the space of annotations being dependent on the application. For example, on CIFAR-10H (Peterson et al., 2019), each expert provides a single label in $[K]$. In contrast, in our dermatology application, using data from (Liu et al., 2020), $B_i$ represents a set of partial rankings whose cardinality depends on $i$. This corresponds to differential diagnoses from dermatologists. However, while

the format of the $B_i$'s can vary, we are always interested in returning prediction sets of classes $C(X) \subseteq [K]$ satisfying a coverage guarantee w.r.t. $\mathbb{P}_{\mathrm{agg}}$ where $\mathbb{P}_{\mathrm{agg}}$ needs to be specified based on the expert annotations $B_i$[2]

We give two simple examples to illustrate how $\mathbb{P}_{\mathrm{agg}}^{Y|X}$ can be obtained and then present a more generic framework. However, the aim of this section is not to provide the best way to aggregate expert opinions. This is application dependent and there is a substantial literature on the topic. Instead, we focus here on showing how such techniques can be exploited in a CP framework:

- **Single labels**: We first revisit and extend the construction presented in Section 1 where $B = (Y^1, ..., Y^p)$ with $Y^q \in [K]$ corresponding to the single label the expert $q$ assigns to $X$. We then consider $\mathbb{P}_{\mathrm{agg}}(Y = y|X, B) = \frac{1}{p} \sum_{k \in [K]} p_k \mathbb{I}(y = k)$ for $p_k = \sum_{q=1}^{p} \mathbb{I}(Y^q = k)$[3]. Note that this approximation is deterministic given $(X, B)$. However, we could also use a bootstrap procedure by resampling the entries of $B$ with replacement.

- **Partial rankings**: In medical applications, it is common for expert annotations to be given by differential diagnoses, corresponding to partial rankings. That is, out of the $K$ possible conditions, each expert returns a partial ranking of conditions because multiple conditions are deemed plausible while a large majority of the conditions can be excluded. While probabilistic models for aggregating such rankings exist (Hajek et al., 2014; Zhu et al., 2023), we follow (Liu et al., 2020) and describe a simple deterministic procedure called inverse rank normalization which we will also exploit in our experiments, see Section 4.1. Let $B = (B^1, ..., B^p)$ be the collection of available partial rankings for data $X$. Each partial ranking $B^q$ is divided into $n_q$ blocks, $B^q = (B_1^q, ..., B_{n_q}^q)$, with $B_i^q \in \subseteq [K]$ and $B_i^q \bigcap B_j^q = \emptyset$ and $B_{n_q}^q$ the collection of excluded conditions. The conditions in $B_i^q$ are assessed as being more plausible than the conditions in $B_j^q$ for $i < j$. We then define

$$\alpha_y = \sum_{q=1}^{p} \sum_{i=1}^{n_q - 1} \frac{1}{i|B_i^q|} \mathbb{I}(y \in B_i^q), \qquad \mathbb{P}_{\mathrm{agg}}(Y = y|X, B) = \frac{\alpha_y}{\sum_{k=1}^{K} \alpha_k}. \tag{6}$$

As above, this approximation is deterministic given $(X, B)$. Again, we could also use a bootstrapping procedure by resampling the entries of $B$ with replacement.

More generally, we assume the distribution of $(X, B)$ admits a density $p(x, b)$. Then, we will denote by $\lambda = (\lambda_1, ..., \lambda_K)$ the probabilities $(\mathbb{P}_{\mathrm{agg}}(Y = 1|X, B), ..., \mathbb{P}_{\mathrm{agg}}(Y = K|X, B))$ obtained by some deterministic or stochastic annotations aggregation procedures. We refer to $\lambda$ as *plausibilities* since they quantify how plausible the different classes are to be the actual ground truth label given the annotations $B$. Then, the resulting marginal density of $(X, \lambda)$ is given by

$$p(x, \lambda) = \int p(x, \lambda, b) \mathrm{d}b = \int p(\lambda|b, x) p(x, b) \mathrm{d}b. \tag{7}$$

In the examples above, note that $p(\lambda|b, x) = p(\lambda|b)$. Then, our aggregation model for label $Y$ can be written as

$$\mathbb{P}_{\mathrm{agg}}(Y = y|X = x) = \int p(y|\lambda) p(\lambda|x) \mathrm{d}\lambda = \int \int \lambda_y p(\lambda|b, x) p(b|x) \mathrm{d}\lambda \mathrm{d}b. \tag{8}$$

In words, $p(b|x)$ corresponds to the annotation process, i.e., how experts draw their annotations when observing data $x$, and $p(\lambda|b, x)$ models aggregation of annotations into plausibilities. Given data $(X, B) \sim p(x, b)$, then $B$ is distributed according to $p(b|X)$ given $X$. So by sampling $\lambda \sim p(\lambda|B, X)$ and then $Y$ such that $\mathbb{P}(Y = y|\lambda) = \lambda_y$, we obtain a sample from $\mathbb{P}_{\mathrm{agg}}(Y = y|X)$. Note however that we cannot obtain i.i.d.

---

[2]Based on this setup, we could in principle develop a CP method returning prediction sets for $B$, e.g., prediction sets of rankings, satisfying a coverage guarantee w.r.t $\mathbb{P}$. However, we are interested here in prediction sets for labels.

[3]$\mathbb{P}_{\mathrm{agg}}(Y = y|X, B)$ obviously converges towards $\mathbb{P}(Y = y|X)$ as $p \to \infty$ if $Y^q \overset{\mathrm{i.i.d.}}{\sim} \mathbb{P}_{\mathrm{agg}}(\cdot|X)$, i.e., this assumes that expert annotations follow the true distribution $\mathbb{P}$.

---

**Algorithm 1 Monte Carlo CP** with $1 - \alpha$ coverage guarantee for $m = 1$ and $1 - 2\alpha$ for $m \geq 2$.

**Input:** Calibration examples $(X_i, \lambda_i)_{i \in [n]}$; test example $X$; confidence level $\alpha$; number of samples $m$
**Output:** Prediction set $C(X)$ for test example

1. Sample $m$ labels $(Y_i^j)_{j \in [m]}$ per calibration example $(X_i)_{i \in [n]}$ where $\mathbb{P}(Y_i^j = k) = \lambda_{ik}$.

2. Calibrate the threshold $\tau$ using

$$\tau = Q\left(\{E(X_i, Y_i^j)\}_{i \in [n], j \in [m]}; \frac{\lfloor \alpha m(n+1) \rfloor - m + 1}{mn}\right). \tag{10}$$

3. Return $C(X) = \{k \in [K] : E(X, k) \geq \tau\}$.

---

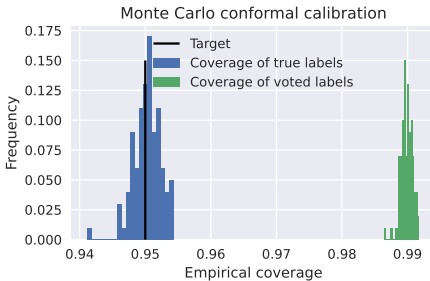

Figure 4: Monte Carlo CP with $m = 10$ sampled labels per calibration example (see Algorithm 1) applied to the toy dataset from Figure 3. We use the true posterior probabilities $\mathbb{P}(Y = y|X = x)$ as plausibilities $\lambda_y$ to sample labels (see Figure 3, left). Again, we show coverage wrt. true or voted labels. In contrast to standard CP calibrated against voted labels, Monte Carlo CP clearly obtains the target coverage (black) of $1 - \alpha = 95\%$ (on average) w.r.t. the true labels.

samples from $\mathbb{P}_{\text{agg}}(Y = y|X)$ as we only have access to one sample from $B$ given $X$. We can only obtain exchangeable samples by repeating sampling from $\lambda \sim p(\lambda|B, X)$ and $\mathbb{P}(Y = y|\lambda) = \lambda_y$. Hence we cannot use concentration inequalities to obtain finite sample bounds for quantities such as $\mathbb{P}_{\text{agg}}(Y \in C(X)|X = x)$ for a given $C(X)$.

We will instead develop CP procedures returning prediction sets satisfying $\mathbb{P}_{\text{agg}}(Y \in C(X)) \geq 1 - \alpha$, it is worth clarifying what this means when $\mathbb{P}_{\text{agg}}(Y|X)$ is ambiguous, i.e., not one-hot. With $\mathbb{P}_{\text{agg}} = \mathbb{P}^X \otimes \mathbb{P}_{\text{agg}}^{Y|X}$, we can rewrite the coverage guarantee as

$$\mathbb{P}_{\text{agg}}(Y \in C(X)) = \mathbb{E}_{X \sim \mathbb{P}^X}\left[\mathbb{E}_{Y \sim \mathbb{P}_{\text{agg}}(\cdot|X)}[\mathbb{I}[Y \in C(X)]]\right]. \tag{9}$$

It makes explicit that, coverage is marginal across examples *and* classes: in ambiguous cases, the prediction set $C(X)$ might cover only part of the classes with non-zero plausibility. We will refer to $\mathbb{P}_{\text{agg}}(Y \in C(X))$ defined in Equation (9) as **aggregated coverage** to emphasize that this is w.r.t. $\mathbb{P}_{\text{agg}}$. This is to contrast from **voted coverage** $\mathbb{P}_{\text{vote}}(Y \in C(X))$ which is w.r.t. $\mathbb{P}_{\text{vote}}$. To obtain prediction sets with aggregated coverage guarantees, we will assume access to a set of exchangeable calibration data $(X_i, \lambda_i)_{i \in [n]}$ of examples and corresponding plausibilities from $p(x, \lambda)$. To obtain this calibration data, we can simply rely on the original (exchangeable) calibration data $(X_i, B_i)_{i \in [n]}$ and then sample $\lambda_i \sim p(\cdot|X_i, B_i)$ where $\lambda_i = (\lambda_{i1}, ..., \lambda_{iK})$. In the examples given above, $\lambda_i$ is given deterministically given $(X_i, B_i)$ so $p(\lambda|x, b)$ is a delta-Dirac mass but, if a bootstrapping procedure is used to account for uncertainty in the annotation process, then it is not anymore.

### 3.2 Introduction to Monte Carlo conformal prediction

We propose Monte Carlo CP, a sampling-based approach to CP under ambiguous ground truth. Given the calibration examples $(X_i, \lambda_i)_{i \in [n]}$, we sample $m$ labels $Y_i^j$ for each $X_i$ according to plausibilities $\lambda_i$, i.e. $\mathbb{P}(Y_i^j = k) = \lambda_{ik}$ and duplicate the corresponding inputs[4]. That is, we obtain $m \cdot n$ new calibration examples $(X_i, Y_i^j)_{i \in [n], j \in [m]}$ and then apply the conformal calibration outlined in Algorithm 1. As shown in Figure 4, Monte Carlo CP overcomes the coverage gap illustrated on our toy dataset in Figure 3.

---

[4]An alternative valid procedure would sample $m$ plausibilities $\lambda_i^j \sim p(\cdot|X_i, B_i)$ for each $X_i$ and then $Y_i^j$ such that $\mathbb{P}(Y_i^j = k) = \lambda_{ik}^j$. When $p(\cdot|X_i, B_i)$ is a delta-mass distribution for all $i$, this coincides with the procedure described previously.

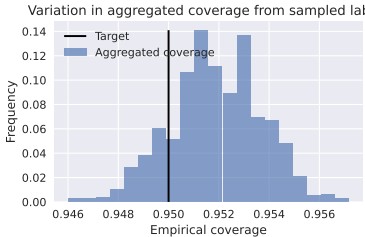 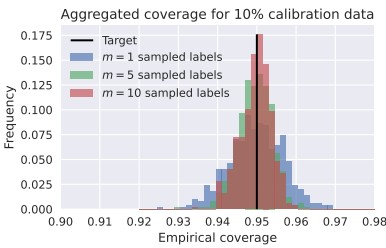 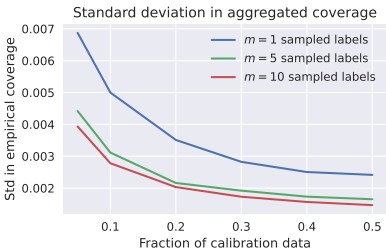

Figure 5: Left: Empirical aggregated coverage for Monte Carlo CP, $m = 1$, with a fixed calibration/test split but different randomly sampled labels $Y_i^j$, cf. Algorithm 1. The approach obtains coverage $1 - \alpha$ across calibration/test splits but overestimates coverage slightly for this particular split. However, in Monte Carlo CP, coverage is marginal across not only test and calibration examples but also the sampled labels during calibration, as shown in this histogram. Middle and right: Instead fixing the sampled labels for different $m$ and plotting variation in coverage across random calibration/test split shows that aggregated coverage is empirically close to $1 - \alpha$ even for $m > 1$. Large $m$ generally reduces the variability in coverage observed across calibration/test splits, especially for small calibration sets (we consider 5% to 50% calibration data).

The aggregated coverage guarantees we will provide for Algorithm 1 are marginal over the sampled labels $(Y_i^j)_{i \in [n], j \in [m]}$. This is illustrated in Figure 5 (left) on the toy dataset of Section 2.2 using a fixed calibration/test split but multiple samples of $(Y_i^j)_{i \in [n], j \in [m]}$. The empirical aggregated coverage across calibration/test splits is $1 - \alpha$ (see Figure 5 (left)) but the variability across such splits is high if the entropy of $\mathbb{P}_{\text{agg}}^{Y|X}$ is high for many calibration data. This variability can be reduced by increasing $m$ (see Figure 5, right), this is especially beneficial in the low calibration data regime.

We now discuss the theoretical coverage properties of Algorithm 1. Recall that we have assumed in Section 3.1 that the joint distribution of $(X_i, B_i)_{i \in [n]}$ and a test data $(X, B)$ is exchangeable. This implies that the joint distribution of $(X_i, Y_i^j)_{i \in [n]}$ and $(X, Y)$ is exchangeable for any $j \in [m]$. For $m = 1$, it is clear that this approach boils down to standard split CP applied to calibration data $(X_i, Y_i^1) \sim \mathbb{P}_{\text{agg}}$ and thus $\mathbb{P}_{\text{agg}}(Y \in C(X)) \geq 1 - \alpha$ follows directly. For $m \geq 2$, the calibration examples include $m$ repetitions of each $X_i$ and exchangeability with the test example $X$, typically used to establish the validity of CP, is not satisfied anymore. Nevertheless, as mentioned earlier, we empirically observe an empirical coverage close to $1 - \alpha$ on average in Figure 5 (middle).

In the following, Section 3.3, we focus on establishing rigorous coverage guarantees for Monte Carlo CP when $m \geq 2$, showing that we can establish $\mathbb{P}_{\text{agg}}(Y \in C(X)) \geq 1 - 2\alpha$. This is akin to the observation in (Barber et al., 2021) that jackknife+, cross-conformal or out-of-bag CP consistently obtain empirical coverage $1 - \alpha$ while they only guarantee $1 - 2\alpha$. In applications where rigorous coverage guarantees better than $1 - 2\alpha$ are necessary, we show in Section 3.4 that this can be achieved by developing an alternative Monte Carlo CP procedure presented in Algorithm 2 which relies on an additional split of the calibration examples.

### 3.3 Coverage $1 - 2\alpha$ by averaging $p$-values

In order to establish the coverage guarantee $\mathbb{P}_{\text{agg}}(Y \in C(X)) \geq 1 - 2\alpha$ when $m \geq 2$ for Algorithm 1, let us introduce for $j \in [m]$:

$$\rho_Y^j = \frac{\sum_{i=1}^{n} \mathbb{I}[E(X_i, Y_i^j) \leq E(X, Y)] + 1}{n + 1}. \tag{11}$$

The random variables $\rho_Y^j$ are $p$-values, i.e. $\mathbb{P}(\rho_Y^j \leq \alpha) \leq \alpha$, since the scores $\{E(X_i, Y_i^j)\}_{i \in [n]}$ and $\{E(X, Y)\}$ are exchangeable for fixed $j \in [m]$. We can now average these quantities over $j \in [m]$ to obtain

$$\bar{\rho}_Y = \frac{1}{m} \sum_{j=1}^{m} \rho_Y^j = \frac{\sum_{j=1}^{m} \left( \sum_{i=1}^{n} \mathbb{I}[E(X_i, Y_i^j) \leq E(X, Y)] + 1 \right)}{m(n + 1)}. \tag{12}$$

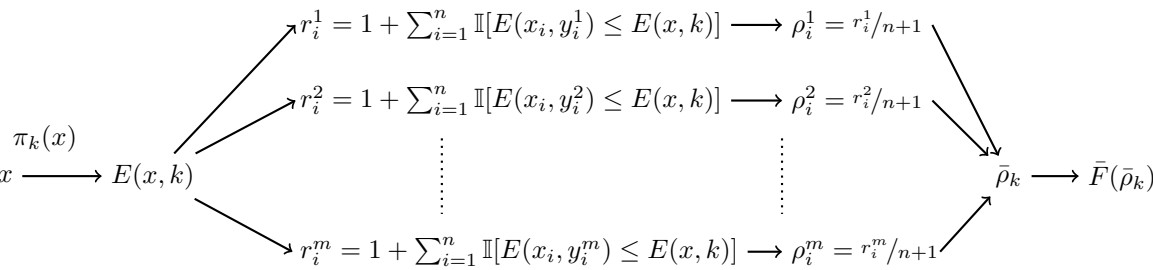

Figure 6: The $m$ label samples result in $(\rho_k^j)_{j\in[m]}$ for the test example $X$, $(\rho_Y^j)_{j\in[m]}$ being p-values. We can average $\rho_k^j$ over $j$ and threshold the resulting average $\bar{\rho}_k$ at level $\alpha$ to obtain the prediction set, this is equivalent to Algorithm 1 as explained in Section 3.3. Alternatively we can combine the $p$-values $\rho_Y^j$ using the ECDF approach described in Section 3.4; see Algorithm 2 for a detailed computational description.

As $\bar{\rho}_Y$ is an average of (dependent) $p$-values, it follows from standard results (Rüschendorf, 1982; Meng, 1994) that $\mathbb{P}(2\bar{\rho}_Y \leq 2\alpha) \leq 2\alpha$, equivalently $\mathbb{P}(\bar{\rho}_Y > \alpha) \geq 1 - 2\alpha$. Hence we have $\mathbb{P}(Y \in C(X)) \geq 1 - 2\alpha$ for $C(X) = \{y \in [K] : \bar{\rho}_y > \alpha\}$. Nevertheless, as discussed previously, we obtain empirical coverage close to $1 - \alpha$, cf. Figure 5 (middle), see also the discussion in (Barber et al., 2021, Section 4). This approach is illustrated in Figure 6. Note that in the context of aggregating $m$ different models, Linusson et al. (2017) also considered similar types of averaging.

Practically, we do not have to compute and actually average $\rho_y^j$ for $j \in [m]$ and $y \in [K]$ to determine $C(X) = \{y \in [K] : \bar{\rho}_y > \alpha\}$. As in Section 2.1, we can reformulate this prediction set as $C(X) = \{k \in [K] : E(X, k) \geq \tau\}$ as in Equation (2) where $\tau$ is the $\lfloor \alpha m(n+1) \rfloor - m + 1$ smallest element of $\{E(X_i, Y_i^j)\}_{i\in[n],j\in[m]}$, equivalently $\tau$ is obtained by computing

$$\tau = Q\left(\{E(X_i, Y_i^j)\}_{i\in[n],j\in[m]}; \frac{\lfloor \alpha m(n+1) \rfloor - m + 1}{mn}\right). \tag{13}$$

and confidence sets are constructed as $C(X) = \{k \in [K] : E(X, k) \geq \tau\}$ for $\tau$ given in Equation (10). This is established in Section B.2 of Appendix B.

## 3.4 Beyond coverage $1 - 2\alpha$

The $1 - 2\alpha$ coverage guarantee from Monte Carlo CP arises from the fact that we use a standard result about average of $p$-values (Rüschendorf, 1982; Meng, 1994). When the $p$-values are independent, a few techniques have been proposed to obtain a $1 - \alpha$ coverage; see e.g. (Cinar & Viechtbauer, 2022) for a comprehensive review. However, in our case, the $p$-values $(\rho_Y^j)_{j\in[m]}$ we want to combine are strongly dependent as they use the same calibration examples $X_i$ and then rely on different pseudo-labels $Y_i^j$ from the same distribution (given by $\lambda_i$). In this setting, many standard methods yield overly conservative results.

We follow here a method that directly estimates the cumulative distribution function (CDF) of the combined, e.g., averaged $p$-values (Balasubramanian et al., 2015; Toccaceli & Gammerman, 2019; Toccaceli, 2019). Let $\bar{\rho}_Y = 1/m \sum_{j=1}^{m} \rho_Y^j$ be the averaged $p$-values. As shown in Figure 7 (left), these averaged $p$-values will not be uniformly distributed. However, if $F$ denotes the CDF of $\bar{\rho}_Y$, then $\rho_Y = F(\bar{\rho}_Y)$ is a $p$-value[5] and thus the prediction set $C(X) = \{y \in [K] : \rho_y > \alpha\}$ will obtain coverage $1 - \alpha$. The true CDF is unknown but we can split the original calibration examples into $X_1, \ldots, X_l$ and $X_{l+1}, \ldots, X_n$ and used the second split to obtained an empirical estimate $\bar{F}$ of $F$. The procedure is described in Algorithm 2.

As $\bar{F}$ is not the true CDF $F$ but an empirical CDF (ECDF) estimate, $C(X) = \{y \in [K] : \bar{F}(\bar{\rho}_y) > \alpha\}$ would only provide an approximate coverage guarantee at level $1 - \alpha$. However, if the original calibration examples $X_{l+1}, \ldots, X_n$ are i.i.d., then the Dvoretzky–Kiefer–Wolfowitz inequality (see e.g. (Wasserman,

---

[5]We have $\mathbb{P}(F(\bar{\rho}_Y) \leq f) = \mathbb{P}(F(F^{-1}(U)) \leq f)$ for $U$ a uniform random variable on $[0,1]$ for $F^{-1}(f) = \inf\{\rho \in \mathbb{R} : F(\rho) \geq f\}$. However $F(F^{-1}(U)) \geq U$ so $\mathbb{P}(F(F^{-1}(U)) \leq f) \leq \mathbb{P}(U \leq f) = f$. Hence, we have $\mathbb{P}(\rho_Y \leq \alpha) \leq \alpha$.

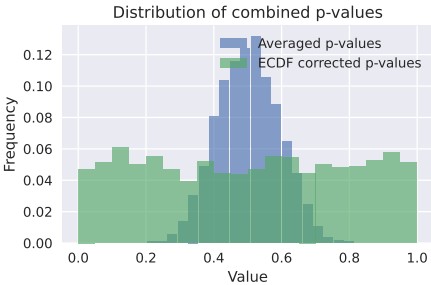 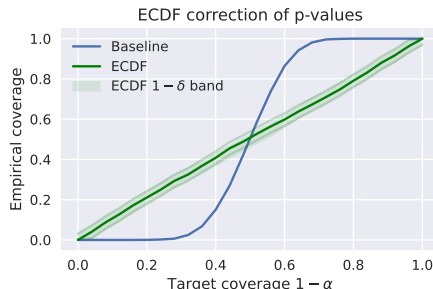

Figure 7: For illustration, we sample 10k $p$-values for $m = 10$ independent tests and duplicate them to obtain $m = 20$ dependent tests. Left: Histograms for averaged and ECDF-corrected $p$-values. The ECDF correction is able to ensure that $p$-values are distributed approximately uniformly. Right: Target confidence level $\alpha$ plotted against the empirical confidence level when calibrating with averaged $p$-values (blue) and the ECDF-corrected ones (green). The Dvoretzky–Kiefer–Wolfowitz inequality provides tight finite-sample guarantees on the ECDF correction, using here $\delta = 0.0001 = 0.01\%$.

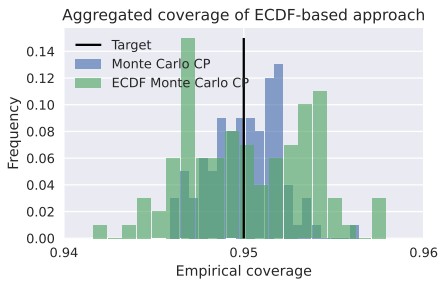

Figure 8: On our toy dataset, we plot aggregated coverage obtained using Monte Carlo CP without and with ECDF correction for $m = 10$ and $l = \lfloor n/2 \rfloor$. As expected, Monte Carlo CP, even for $m \geq 2$, does not result in reduced coverage. Thus, ECDF correction does not yield meaningfully different results besides exhibiting higher variation in coverage across calibration/test splits due to the additional split $l$.

2006)) provides rigorous finite sample guarantees for the ECDF, i.e.

$$\mathbb{P}(\sup_{f \in [0,1]} |\bar{F}(f) - F(f)| > \epsilon) \leq 2 \exp(-2(n-l)\epsilon^2). \tag{14}$$

Basic manipulation results in a confidence band for $\bar{F}$

$$\mathbb{P}(\forall f \in [0,1], \bar{F}_-(f) \leq F(f) \leq \bar{F}_+(f)) \geq 1 - \delta \quad \text{with} \quad \bar{F}_{\pm}(f) = \frac{\min}{\max} \left\{ \bar{F}(f) \pm \sqrt{\frac{1}{2(n-l)} \log \frac{2}{\delta}}, \frac{1}{0} \right\}. \tag{15}$$

This confidence band is illustrated shown in Figure 7 (right). We can now define the prediction set $C(X) = \{y \in [K] : \rho_y > \alpha\}$ for $\rho_Y = \bar{F}_+(\bar{\rho}_Y)$ and show that it satisfies $\mathbb{P}_{\text{agg}}(Y \in C(X)) \geq (1 - \alpha)(1 - \delta)$. Indeed, writing $\bar{F}_+ \geq F$ to abbreviate $\bar{F}_+(f) \geq F(f) \, \forall f$, we have $\mathbb{P}(\bar{F}_+(\bar{\rho}_Y) > \alpha) \geq \mathbb{P}(\bar{F}_+(\bar{\rho}_Y) > \alpha | \bar{F}_+ \geq F)\mathbb{P}(\bar{F}_+ \geq F) \geq \mathbb{P}(F(\bar{\rho}_Y) > \alpha | \bar{F}_+ \geq F)\mathbb{P}(\bar{F}_+ \geq F)$. Now we have from Equation (14) that $\mathbb{P}(\bar{F}_+ \geq F) \geq 1 - \delta$ and $\mathbb{P}(F(\bar{\rho}_Y) > \alpha | \bar{F}_+ \geq F) = \mathbb{P}(F(\bar{\rho}_Y) > \alpha)$ as the probability in Equation (14) is over $(X_i, Y_i)_{i \in \{l+1,\ldots,n\}}$ while $\bar{\rho}_Y$ is only a function of $(X_i, Y_i^j)_{i \in [l], j \in [m]}$ and $(X, Y)$. Using the fact that $\mathbb{P}(F(\bar{\rho}_Y) > \alpha) \geq 1 - \alpha$, we obtain the desired coverage guarantee.

Figure 8 compares the empirical coverage obtained using standard CP with true labels to both approaches of Monte Carlo CP (cf. Algorithms 1 and 2) to verify that this approach indeed obtains coverage $1 - \alpha \approx (1 - \alpha)(1 - \delta)$ on average for $\delta \ll 1$. As discussed, without the ECDF correction discussed here, this is only empirical as the approach discussed in the previous section can only guarantee $1 - 2\alpha$. We also did not observe a significant difference in prediction set size when using or not using the ECDF correction.

## 3.5 Summary

We have presented two Monte Carlo CP techniques, Algorithms 1 and 2, which rely on sampling multiple synthetic pseudo-labels from $\mathbb{P}_{\text{agg}}^{Y|X}$ for each calibration data. In this setting, the coverage guarantees we

---

**Algorithm 2 ECDF Monte Carlo CP** with $(1-\alpha)(1-\delta)$ coverage guarantee.

---

**Input:** Calibration examples $(X_i, \lambda_i)_{i\in[n]}$; test example $X$; confidence levels $\alpha, \delta$; data split $1 \leq l \leq n-1$; number of samples $m$

**Output:** Prediction set $C(X)$ for test example $X$

1. Sample $m$ labels $(Y_i^j)_{j\in[m]}$ per calibration example $(X_i)_{i\in[l]}$ where $\mathbb{P}(Y_i^j = k) = \lambda_{ik}$.

2. Sample one label $Y_i$ per calibration example $(X_i)_{i\in\{l+1,\dots,n\}}$ where $\mathbb{P}(Y_i = k) = \lambda_{ik}$.

3. Compute $(\bar{\rho}^i)_{i\in\{l+1,\dots,n\}}$ where

$$\bar{\rho}^i = \frac{\sum_{j=1}^m \left( \sum_{p=1}^l \mathbb{I}[E(X_p, Y_p^j) \leq E(X_i, Y_i)] + 1 \right)}{m(l+1)}. \tag{16}$$

4. Build the ECDF $\bar{F}(f) = \frac{1}{n-l} \sum_{i=l+1}^n \mathbb{I}[\bar{\rho}^i \leq f]$ and its upper bound $\bar{F}_+(f)$ using Equation (15).

5. For test example $X$, compute for $k \in [K]$

$$\bar{\rho}_k = \frac{\sum_{j=1}^m \left( \sum_{p=1}^l \mathbb{I}[E(X_p, Y_p^j) \leq E(X, k)] + 1 \right)}{m(l+1)}, \quad \bar{\rho}_k^{\text{corr}} = \bar{F}_+(\bar{\rho}_k). \tag{17}$$

6. Return $C(X) = \{k \in [K] : \bar{\rho}_k^{\text{corr}} > \alpha\}$.

---

obtain have to be understood marginally across calibration examples *and* their sampled labels. We summarize the advantages and drawbacks of these methods in Table 1. In the following, we show that these developments are relevant beyond the setting of ambiguous ground truth and discuss related work.

| Procedure | Theoretical coverage | Avg. empirical coverage | Variability emp. coverage | Additional calibration split |
|---|---|---|---|---|
| Alg. 1, $m = 1$ | $1 - \alpha$ | $1 - \alpha$ | High | No |
| Alg. 1, $m \geq 2$ | $1 - 2\alpha$ | $1 - \alpha$ | Low | No |
| Alg. 2 | $(1-\alpha)(1-\delta)$ | $(1-\alpha)(1-\delta)$ | Low | Yes |

Table 1: Overview of the proposed Monte Carlo CP procedures in terms of the provided theoretical coverage guarantee, the observed empirical coverage (averaged across sampled pseudo labels), variability of the empirical coverage (w.r.t. sampled pseudo labels) and whether an additional split of the calibration data is required.

## 3.6 Applications to related problems

**Multi-label classification:** Let $\mathcal{Y}_i \subseteq [K]$ be the known, ground truth multi-label *set* for each calibration example $X_i$. For simplicity, we express the multi-label setting using plausibilities $\lambda_i$ that divide probability mass equally across the $L_i = |\mathcal{Y}_i| \geq 1$ labels. We can then apply Monte Carlo CP to this setup. When $m \gg 1$, we obtain calibration examples where the proportion of $(Y_i^j)_{j\in[m]}$ equal to a given class is approximately $1/L_i$. This is related to an empirical existing method to perform multi-label CP (Tsoumakas & Katakis, 2007; Wang et al., 2014; 2015) where for each calibration example $(X_i, \mathcal{Y}_i)$ we perform CP using the calibration examples $(X_i, Y_i^1), \dots, (X_i, Y_i^{L_i})$. Monte Carlo CP provides coverage guarantees on $\mathbb{P}_{\text{agg}}(Y \in C(X))$ for

$$\mathbb{P}_{\text{agg}}(Y = y | X = x) = \sum_{\mathcal{Y}} \lambda(\mathcal{Y})_y \, p(\mathcal{Y}|x) \tag{18}$$

with $p(\mathcal{Y}|x)$ being the conditional probability of the set $\mathcal{Y}$ given $X = x$ and $\lambda(\mathcal{Y})_y = \mathbb{I}(y \in \mathcal{Y})/|\mathcal{Y}|$.

**Data augmentation and robustness:** Consider a scenario where we have exchangeable calibration data $(X_i, Y_i)_{i \in [n]}$ for $Y_i \in [K]$. We want to augment the set of calibration data by using data augmentation, i.e. for each $X_i^1 := X_i$ we sample additional $X_i^2, ..., X_i^m \sim p(\cdot | X_i)$. For example, these could correspond to versions of $X_i$ which are (adversarially or randomly) perturbed or corrupted, rotated, flipped, etc. As we usually train with data augmentation and frequently want to improve robustness against distribution shifts or specific types of perturbations, considering these augmentations for calibration is desirable. However, when using the augmented set $m \cdot n$ of calibration data $(X_i^j, Y_i)_{i \in [n], j \in [m]}$, the joint distribution of calibration data and test data is not exchangeable anymore. However, we can still provide rigorous coverage guarantees using a procedure very similar to Monte Carlo CP. For each test data $(X, Y)$, we set $X^1 = X$ and sample augmentations $X^2, ..., X^m \sim p(\cdot | X)$. We then proceed by averaging the following $m$ p-values

$$\rho_Y^j = \frac{\sum_{i=1}^n \mathbb{I}[E(X_i^j, Y_i) \leq E(X^j, Y)] + 1}{n + 1}, \quad \bar{\rho}_Y = \frac{1}{m} \sum_{j=1}^m \rho_Y^j. \tag{19}$$

By following arguments similar to Section 3.3, the prediction set $C(X^1, X^2, ..., X^n) = \{y \in [K] : \bar{\rho}_y > \alpha\}$ satisfies $\mathbb{P}_{\text{aug}}(Y \in C(X^1, X^2, ..., X^n)) \geq 1 - 2\alpha$ where $\mathbb{P}_{\text{aug}}$ is the joint distribution of the test data $(X = X^1, Y)$ and the augmentations $(X^2, .., X^m)$. In this case, the coverage guarantee is marginal across $(X^1, ..., X^2, Y)$ and $(X_i^j, Y_i)_{i \in [n], j \in [m]}$.

### 3.7 Related work

CP (Vovk et al., 2005) has recently found numerous applications in machine learning, see e.g. (Romano et al., 2019; Sadinle et al., 2019; Romano et al., 2020; Angelopoulos et al., 2021; Stutz et al., 2021; Fisch et al., 2022). In this paper, we focus on split CP (Papadopoulos et al., 2002). However, there are also transductive and cross-validation/bagging-inspired variants being studied (Vovk et al., 2005; Vovk, 2015; Steinberger & Leeb, 2016; Barber et al., 2021; Linusson et al., 2020). Our work is related to these approaches in that many of them guarantee coverage $1 - 2\alpha$ while empirically obtaining coverage close to $1 - \alpha$. For example, cross-CP (Vovk, 2015) was recently shown to satisfy a $1 - 2\alpha$ guarantee in (Vovk et al., 2018; Kim et al., 2020). Moreover, this guarantee is also based on combining $p$-values without making any assumption about their dependence structure.

As outlined before, our work is also related to CP for multi-label classification which faces similar challenges as CP for ambiguous ground truth (Wang et al., 2014; 2015; Lambrou & Papadopoulos, 2016; Papadopoulos, 2014; Cauchois et al., 2021). Finally, our work has similarities to work on adversarially robust CP (Gendler et al., 2022), especially in terms of our ideal coverage guarantee.

There is also a long history of work on combining dependent or independent $p$-values (Fisher, 1925; Rüschendorf, 1982; Meng, 1994; Heard & Rubin-Delanchy, 2017). Key work has been done in (Vovk et al., 2018), showing results without dependence assumption and thereby establishing guarantees for, e.g., cross-CP. Similar to us, (Balasubramanian et al., 2015; Linusson et al., 2017; Toccaceli & Gammerman, 2019; Toccaceli, 2019) use the ECDF to combine $p$-values but they do not provide rigorous coverage guarantees for this procedure.

## 4 Applications

### 4.1 Main case study: skin condition classification

In the main case study of this paper, we follow (Liu et al., 2020; Stutz et al., 2023) and consider a very ambiguous as well as safety-critical application in dermatology: skin condition classification from multiple images. We use the dataset of Liu et al. (2020) consisting of 1949 test examples and 419 classes with up to 6 color images resized to $448 \times 448$ pixels. The classes, i.e., conditions, were annotated by various dermatologists who provide partial rankings. These rankings are aggregated deterministically to obtain the plausibilities $\lambda$ using the inverse rank normalization procedure of (Liu et al., 2020) described in Section 3.1. We followed (Roy et al., 2022; Stutz et al., 2023) to train a classifier that achieves 72.6% top-3 accuracy

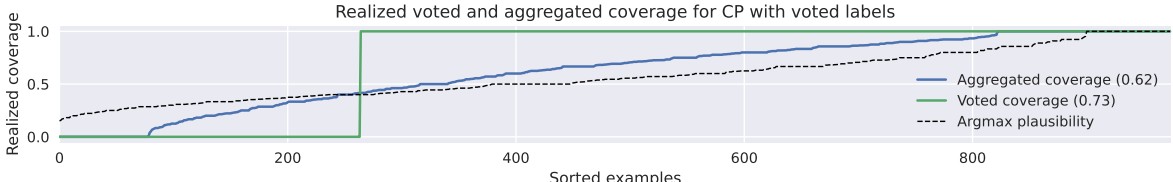

Figure 9: Realized voted coverage, i.e., $\mathbb{I}[\arg\max_k \lambda_{ik} \in C(X_i)]$ (green), and aggregated coverage, i.e., $\sum_{y\in[K]} \lambda_{iy}\mathbb{I}[y \in C(X_i)]$ (blue), for standard CP using voted labels, i.e., $\arg\max_k \lambda_{ik}$. Additionally, we plot the maximum plausibility per example (dashed black) as proxy of ambiguity. We sort examples on the x-axis according to each plot individually. Clearly, many cases are very ambiguous and aggregated coverage is underestimated severely (62% vs. the target of 73%).

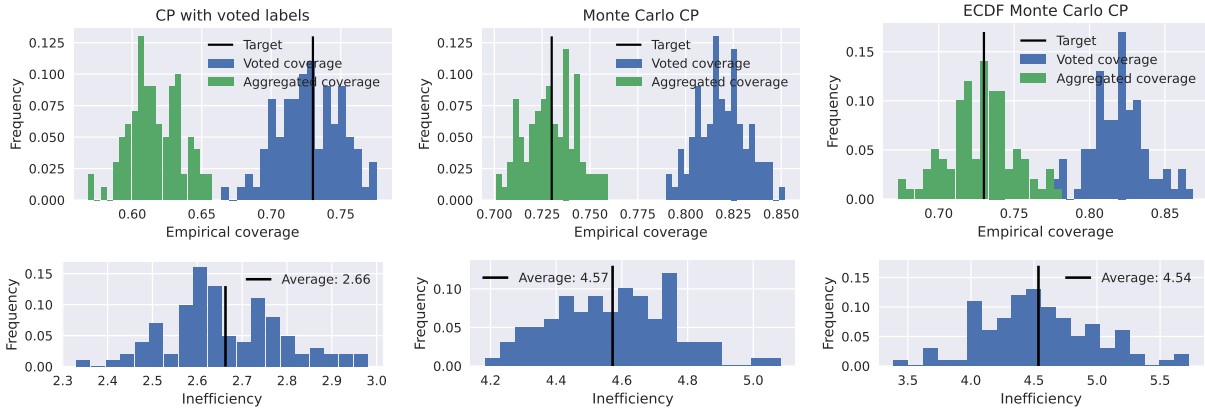

Figure 10: Comparison between CP applied to voted labels (left), Monte Carlo CP (middle) and ECDF Monte Carlo CP (right) in terms of voted coverage (blue) and aggregated coverage (green) across 100 random calibration/test splits (top) as well as inefficiency (bottom). We use $m = 10$. With voted labels does not reach the 73% aggregated target coverage (black). Monte Carlo CP overcomes this gap at the expense of higher inefficiency (bottom). Using ECDF-corrected $p$-values increases observed variation slightly due to the additional calibration data split (half of the original split).

against the voted label from the plausibilities[6]. We chose a coverage level of $1-\alpha = 73\%$ for our experiments (with results for $\alpha = 0.1$ in the appendix) to stay comparable to the base model.

In Figure 9, we highlight how ambiguous the plausibilities for skin condition classification are: in dotted black, we plot the largest plausibility against (sorted) examples. As baseline, we performed CP using the classifier's softmax output as conformity scores and calibrating against the voted labels $\hat{y} := \arg\max_k \lambda_{ik}$ per calibration example $(X_i, \lambda_i)$. In blue, we plot the realized coverage by evaluating $\mathbb{I}[\hat{y} \in C(X_i)]$ per example. This is a step function and roughly 27% of the examples on the x-axis are covered. In green, we plot the realized aggregated coverage by evaluating $\sum_{k\in[K]} \lambda_{ik}\mathbb{I}[k \in C(X_i)]$ per example. For many examples, aggregated coverage lies in between $(0,1)$ showing that the obtained prediction sets only cover part of the plausibility mass. More importantly, aggregated coverage is 62% on average, i.e., significantly under-estimated by calibrating against voted labels. This is the core problem we intend to address.

While the above results consider a fixed calibration/test split, Figure 10 shows our overall results across 100 random splits. As suggested above, on the left, we show how standard CP applied to voted labels does not achieve (as expected) the target of 73% for the aggregated coverage $\mathbb{P}_{\text{agg}}(Y \in C(X))$ (green). As highlighted in Figure 11, the prediction sets miss highly plausible conditions. In the middle and on the right, we demonstrate that Monte Carlo CP with $m = 10$ overcomes this problem and achieves (on average) the target aggregated coverage of 73%. Note that coverage w.r.t. the voted label (blue) increases alongside

---

[6]To be precise, in 72.6% of the cases, the voted label from the plausibilities is included in the top-3 prediction set derived from the predicted softmax output $\pi(x)$ *without* any conformal calibration.

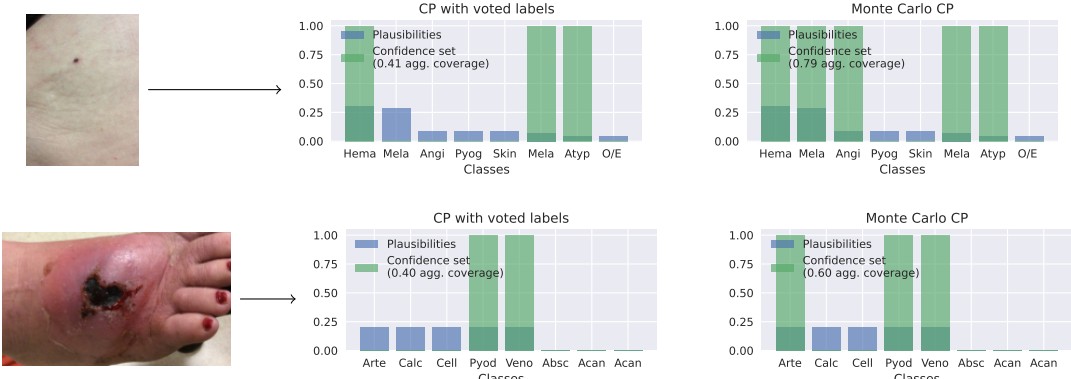

Figure 11: Comparison of CP with voted labels and Monte Carlo CP on two concrete examples. Both are ambiguous cases as shown by the high-entropy plausibilities. Monte Carlo CP clearly covers more plausibility mass (i.e., yields higher realized aggregated coverage), potentially improving patient outcome. Appendix C includes more qualitative results.

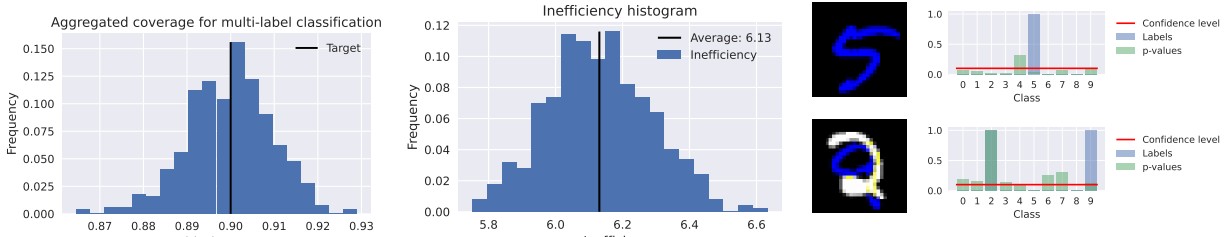

Figure 12: The multi-label CP strategy of Wang et al. (2015; 2014); Tsoumakas & Katakis (2007) is a slight variant of our Monte Carlo CP approach. As shown on this example of up to two overlaid, differently colored digits, Monte Carlo CP achieves target coverage of 90% (top left). However, it is free to decide how many labels to cover per examples (top middle) and, due to the poor performance of the base model (58.8% aggregated coverage), inefficiency is rather high. On the bottom, we show two examples of our dataset with the corresponding ground truth label set $\mathcal{Y}$ (blue) and the obtained Monte Carlo $p$-values (green).

aggregated coverage (i.e., the gap between voted and aggregated coverage remains approximately the same). In terms of inefficiency, avoiding over-confidence by improving aggregated coverage leads to a significant increase in the average prediction set size from 2.66 to 4.57. However, Figure 11 highlights that this is necessary for the prediction sets to include relevant conditions.

## 4.2 Case study: multi-label classification

In Figure 12 we consider a simple MNIST-based multi-label classification problem with up to two, differently colored digits per image. We trained 10 multi-layer perceptrons with 100 hidden units for each digit to determine if the digit is present in the image. This simple classifier achieves 58.8% aggregated coverage when thresholding the 10 individual sigmoids at 0.5. As discussed in Section 3.6, a common strategy (Wang et al., 2015; 2014; Tsoumakas & Katakis, 2007) of performing multi-label CP consists in repeating each example according to its number of labels (here, at most 2). We can achieve something similar with rigorous aggregated coverage guarantee by uniformly sampling labels to perform Monte Carlo CP. We illustrate this procedure in Figure 12 (top left) where we show that it achieves the 90% coverage target. Furthermore, our discussion in Section 3.2 establishes the corresponding guarantee of $1 - 2\alpha$ without ECDF correction. However, it is important to understand what aggregated coverage means for multi-label classification: CP decides how many of the labels it intends to cover per example in order to achieve the marginal coverage guarantee. This is highlighted in Figure 12 (bottom) showing the corresponding $p$-values for two examples. In the first example, only one out of two examples is covered by the prediction set.

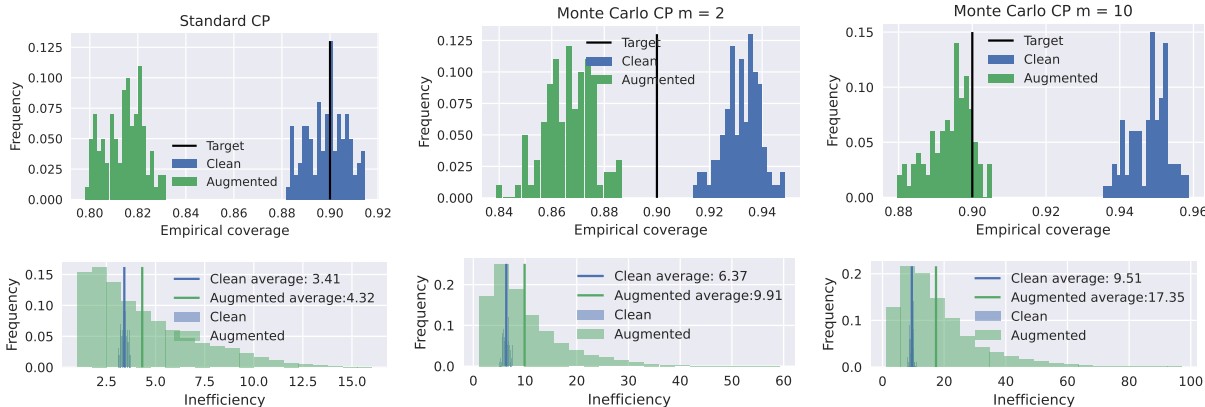

Figure 13: Standard CP applied to original images (left) and Monte Carlo CP (middle and right) applied to the original plus augmented images from ImageNet for $m = 2$ (middle) and $m = 10$ (left). We show coverage on original (blue) and augmented images (green, average across 25 AutoAugment augmentations per image). Monte Carlo CP is able to increase coverage on augmented images significantly at the expense of higher inefficiency (on both original and augmented images). Using more augmentations during calibration generally increases coverage on augmented images and reduces the observed variation across random calibration/test splits.

### 4.3 Case study: data augmentation and robustness

We also apply Monte Carlo CP in the data augmentation and robustness setting outlined Section 3.6. Specifically, we took a pre-trained MobileNet V2 (Howard et al., 2017) achieving 71.3% (top-1) accuracy on the first 5k test examples of ImageNet (Russakovsky et al., 2015) and additionally evaluated it on augmented images using AutoAugment (Cubuk et al., 2018). We generated 25 random augmentations per test example. The model achieves 60.2% accuracy on average, significantly lower than on the original images. Similarly, Figure 13 shows a significant gap in coverage when calibrating only on the original images. While coverage on original examples is around the target of 90%, depending on the random calibration/test split, coverage of augmented images is only slightly above 80%. With the Monte Carlo CP procedure described in Section 3.6, we can use as calibration data the original *and* augmented images. This procedure increases coverage on the augmented images significantly at the cost of higher inefficiency. As coverage is marginal and thus split across augmented and original images, the coverage increase is largest when using more augmented images during calibration. We interpret these results in two ways: First, there is no reason anymore to train state-of-the-art models with data augmentation but discard augmented images during calibration. Second, our Monte Carlo CP approach is effective in improving robustness against augmentations or other corruptions.

## 5 Discussion

In many classification tasks, the available ground truth labels arise from a *voting* process relying on several expert annotations resulting in a one-hot distribution $\mathbb{P}^{Y|X}_{\text{vote}}$. However, in scenarios where annotators tend to disagree because ground truth labels are ambiguous, this voting approach ignores the label uncertainty. Thus, performing CP with such voted labels can only guarantee coverage w.r.t. $\mathbb{P}^{Y|X}_{\text{vote}}$. This can have severe consequences, especially in safety-critical applications such as the dermatology case study in this paper. Instead, we use standard procedures from the literature to *aggregate* expert opinions and return a non-degenerate conditional probability distribution $\mathbb{P}^{Y|X}_{\text{agg}}$ that can capture uncertainty about $Y$. In this paper, we proposed two Monte Carlo CP procedures which can output prediction sets satisfying some pre-specified coverage guarantees under the corresponding distribution $\mathbb{P}_{\text{agg}} = \mathbb{P}^X \otimes \mathbb{P}^{Y|X}_{\text{agg}}$ by sampling $m$ exchangeable synthetic labels from $\mathbb{P}^{Y|X}_{\text{agg}}$ for each calibration data. This allows to reduce the variability of the empirical coverage across realizations of the sampled labels. For a test example $X$, the prediction set outputted by such procedures can be thought as the one expert would assign if they had access to $X$ and their annotations

were aggregated using $\mathbb{P}_{\text{agg}}^{Y|X}$. In the scenario considered here where true labels are never observed, this is the best one can hope for. We have established rigorous coverage guarantees for these Monte Carlo CP procedures despite the fact that the joint distribution between the calibration data and the test data is not exchangeable. While the averaged empirical coverage provided by Algorithm 1 is $1 - \alpha$, our theoretical result only guarantees $1 - 2\alpha$. In applications where rigorous tighter theoretical guarantees are required, we show how it is possible to modify this procedure to obtain $(1 - \alpha)(1 - \delta)$ coverage at the cost of an additional calibration split, see Algorithm 2. We demonstrated the applicability of these approaches in the safety-critical and particularly ambiguous setting of skin condition classification. In this context, the use of voted labels leads to overconfident prediction sets which leads to severe under-coverage of critical conditions. Our Monte Carlo CP overcomes this gap empirically and theoretically. We also demonstrated how our methodology allows conformal calibration with augmented examples and provides, for the first time, a coverage guarantee for multi-label CP.

We also want to highlight the assumptions that underlie our work and some potential extensions. First, we assume access to calibration data of the form $(X_i, B_i)_{i \in [n]}$ for annotations $B_i$ from which we obtain calibration data $(X_i, \lambda_i)_{i \in [n]}$ for plausibilities $\lambda_i$ and then, by sampling, pseudo-labels $(X_i, Y_i^j)_{i \in [n], j \in [m]}$. While assuming that the joint distribution of $(X_i, Y_i^1)_{i \in [n]}$ and test data $(X, Y)$ is exchangeable for Algorithm 1 and that these data are i.i.d. for Algorithm 2 is standard, this is not true in presence of distribution shift at test time. However, we believe it should be possible to adapt Monte Carlo CP to this setting using the techniques developed in (Tibshirani et al., 2019). We note finally that it is also possible to bypass having to sample pseudo-labels and perform conformal calibration directly on the plausibilities using calibration data $(X_i, \lambda_i)_{i \in [n]}$ but this leads to prediction intervals which are difficult to interpret and exploit. Second, our method relies on an aggregation model $\mathbb{P}_{\text{agg}}^{Y|X}$. However, we emphasize that we are not making more assumptions than is currently made by applying CP to voted labels. On the contrary, we explicitly state the typically implied assumptions regarding label collection. Finally, it is important to keep in mind that, as any CP technique, the proposed Monte Carlo CP procedures only provide unconditional coverage guarantees of the form $\mathbb{P}_{\text{agg}}(Y \in C(X)) \geq 1 - \alpha$ and not guarantees of the form $\mathbb{P}_{\text{agg}}(Y \in C(X)|X = x) \geq 1 - \alpha$.

### Broader impact

Our work addresses the problem of conformal calibration in ambiguous settings where ground truth labels are not crisp as generally implicitly assumed in supervised machine learning. As discussed in our skin condition classification case study, ignoring this ambiguity can have severe consequences if used standard techniques relying on voted labels: key conditions (such as the cancerous "Melanoma" in Figure 11) may not be covered in the outputted prediction sets despite experts including them in their annotations. This can have very immediate negative consequences for patients as well as increase cost and strain on the healthcare system. Therefore, we generally view the impact of our work very positively, especially in the deployment of safety-critical applications such as in health. However, it is also important to look into how the methods proposed here affect different demographic groups.

## Data availability

The de-identified dermatology data used in this paper is not publicly available due to restrictions in the data-sharing agreements.

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

# A   Annotations in popular datasets

Many popular machine learning datasets have been created using annotators to determine labels. Often, they also use very basic aggregation and voting mechanisms (corresponding to $\mathbb{P}_{\text{agg}}$ and $\mathbb{P}_{\text{vote}}$) such as majority voting or averaging. For example, the list of most cited datasets on Papers with Code[7] includes the following:

- The CIFAR10 dataset (Krizhevsky, 2009) is a popular multiclass classification benchmark of $32 \times 32$ pixel color images. It was labeled using a strategy similar to majority voting: first, a crowd-sourced label was collected which was then verified and potentially corrected by an author.

- CIFAR10H (Peterson et al., 2019) revisited the CIFAR10 dataset by gathering additional human annotations where each annotator provides a single label.

- ImageNet (Russakovsky et al., 2015) uses multiple annotators with majority voted labels.

- COCO (Lin et al., 2014) labels a category as present if any out of 8 annotators labeled it as present (even if all other annotators disagree).

- The Stanford sentiment treebank (Socher et al., 2013) includes 3 annotations per example which are used in various ways during evaluation.

- MultiNLI (Williams et al., 2018) includes 5 annotations which are majority voted.

- The semantic textual similarity benchmark (Cer et al., 2017) averages scores across multiple annotators.

- The Recognizing Textual Entailment datasets[8] also consider multiple annotators and several editions of the corresponding challenge simply neglected examples with disagreeing annotators (equivalent to majority voting while ignoring ambiguous examples).

The above natural language processing datasets are all part of the GLUE benchmark (Wang et al., 2019) – the go-to benchmark for natural language understanding. Datasets beyond (multi-label) classification also often include multiple annotations. VQA (Goyal et al., 2017), for example, includes 10 free-form answers per question; aggregating them is clearly non-trivial. For many datasets, it might also be unclear how annotations have been collected or aggregated. The Quora Question Pairs webpage[9] explicitly mentions label errors but does not give details on annotation and aggregation. All of these examples emphasize that aggregating annotations and using majority voted or averaged labels is extremely common across supervised machine learning. This means that plausibilities are typically readily obtainable and the core problem we address – using majority voted labels on ambiguous tasks for calibration leads to under-coverage – is highly relevant.

# B   Calibration threshold and $p$-values

### B.1   Single p-value

We include for completeness a proof of the results presented in Section 2.1. These are standard results. The p-value formulation detailed here will be then extended to establish the validity of some of the procedures proposed in this work.

We first establish that the prediction set defined by Equations (2) and (3) satisfies

$$1 - \alpha \le \mathbb{P}(Y \in C(X)) \le 1 - \alpha + \frac{1}{n+1}, \tag{20}$$

the upper bound requiring the additional assumption that the conformity scores are almost surely distinct.

---

[7]https://paperswithcode.com/datasets?q=&v=lst&o=cited
[8]https://tac.nist.gov//2008/rte/past_data/index.html
[9]https://quoradata.quora.com/First-Quora-Dataset-Release-Question-Pairs

Let us write $(X_{n+1}, Y_{n+1}) = (X, Y)$ and $S_i = E(X_i, Y_i)$. As $(S_i)_{i \in [n+1]}$ are exchangeable, then a direct application of (Romano et al., 2019, Appendix A, Lemma 2) shows that for any $\alpha \in (0,1)$ and $\tau' = Q\left(\{-S_i\}_{i \in [n]}; \lceil (1-\alpha)(n+1) \rceil / n \right)$

$$\mathbb{P}(-S_{n+1} \leq \tau') \geq 1 - \alpha \tag{21}$$

and, if the random variables $(S_i)_{i \in [n+1]}$ are almost surely distinct, then

$$\mathbb{P}(-S_{n+1} \leq \tau') \leq 1 - \alpha + \frac{1}{n+1}. \tag{22}$$

So $C(X_{n+1})$ includes all the values $y$ such that $-S(X_{n+1}, y)$ is smaller or equal than the $\lceil (1-\alpha)(n+1) \rceil$ smallest values of $(-S_i)_{i \in [n]}$. This is equivalent to considering all the values $y$ such that $S(X_{n+1}, y)$ is larger or equal than the $\lfloor \alpha(n+1) \rfloor$ smallest values of $(S_i)_{i \in [n]}$. Hence this corresponds to the prediction set defined by Equations (2) and (3).

We now show that this prediction set can also be obtained by thresholding $p$-values. Indeed Equation (4) can be rewritten as

$$\rho_{Y_{n+1}} = \frac{\sum_{i=1}^{n+1} \mathbb{I}(S_i \leq S_{n+1})}{n+1}. \tag{23}$$

As $(S_i)_{i \in [n+1]}$ are exchangeable, then $\rho_{Y_{n+1}}$ is uniformly distributed on $\{\frac{1}{n+1}, \frac{2}{n+1}, ..., 1\}$ if the distribution of the scores is continuous and thus $\rho_{Y_{n+1}}$ is a $p$-value, i.e. $\mathbb{P}(\rho_{Y_{n+1}} \leq \alpha) \leq \alpha \iff \mathbb{P}(\rho_{Y_{n+1}} > \alpha) \geq 1 - \alpha$. It can be checked that this property still holds if the distribution of the scores is not continuous (see e.g. (Bates et al., 2023)). So if we define the prediction set as

$$C(X_{n+1}) = \{y : \rho_y > \alpha\} \tag{24}$$

then by construction it follows that

$$\mathbb{P}(Y_{n+1} \in C(X_{n+1})) \geq 1 - \alpha. \tag{25}$$

We show here that Equation (24) does indeed coincide with the set defined by Equations (2) and (3). We have

$$\rho_y > \alpha \iff \sum_{i=1}^{n} \mathbb{I}(S_i \leq E(X_{n+1}, y)) > \alpha(n+1) - 1. \tag{26}$$

If $\alpha(n+1)$ is an integer then $\sum_{i=1}^{n} \mathbb{I}(S_i \leq E(X_{n+1}, y)) \geq \alpha(n+1)$, i.e. $E(X_{n+1}, y)$ is larger or equal than the $\alpha(n+1)$ smallest values of $(S_i)_{i \in [n]}$. If $\alpha(n+1)$ is not an integer then it means that $\sum_{i=1}^{n} \mathbb{I}(S_i \leq E(X_{n+1}, y)) \geq \lceil \alpha(n+1) \rceil - 1$, i.e. $E(X_{n+1}, y)$ is larger than the $\lceil \alpha(n+1) \rceil - 1$ smallest values of $(S_i)_{i \in [n]}$. However, we have $\lceil \alpha(n+1) \rceil - 1 = \lfloor \alpha(n+1) \rfloor$ as $\alpha(n+1)$ is not an integer. So overall, we have that $\rho_y > \alpha$ corresponds to $E(X_{n+1}, y)$ being larger or equal than the $\lfloor \alpha(n+1) \rfloor$ smallest values $(S_i)_{i \in [n]}$. So the prediction set in Equation (24) does coincide with the set defined by Equations (2) and (3).

Finally note that when the distributions of the scores is continuous, i.e. the scores are almost surely distinct, then we also have $\mathbb{P}(\rho_{Y_{n+1}} \leq \alpha) \geq \alpha - 1/n+1$ so $\mathbb{P}(\rho_{Y_{n+1}} > \alpha) \leq 1 - \alpha + 1/n+1$.

## B.2 Average of p-values

We establish here the expression of the prediction set obtained by thresholding the following average $p$-value

$$\rho_{Y_{n+1}} = \frac{\sum_{j=1}^{m} \sum_{i=1}^{n+1} \mathbb{I}(S_i^j \leq S_{n+1}^j)}{m(n+1)} \tag{27}$$

where $S_{n+1}^j = S_{n+1}$ for all $j \in [m]$. We know that by construction the set $C(X_{n+1}) = \{y : \rho_y > \alpha\}$ is such that

$$\mathbb{P}(Y_{n+1} \in C(X_{n+1})) \geq 1 - 2\alpha. \tag{28}$$

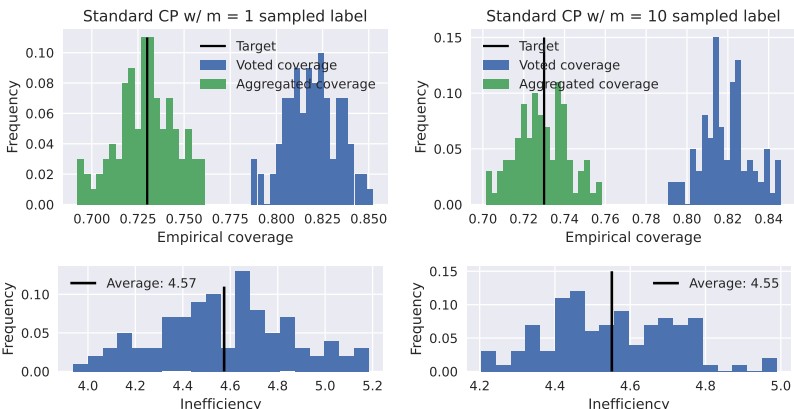

Figure 14: Complementary to Figure 10, we show results for standard CP with $m = 1$ and $m = 10$ sampled labels per example in terms of voted coverage (blue) and aggregated coverage (green) across 100 random calibration/test splits (top) as well as inefficiency (bottom). As for Monte Carlo CP in Algorithm 2, we sample $m$ labels from the plausibilities, but use the quantile from Equation (3) for calibration. For $m = 1$, this coincides with Monte Carlo CP; for $m > 1$, however, the quantile computation differs from Equation (13).

We have

$$\rho_y > \alpha \iff \sum_{j=1}^{m} \sum_{i=1}^{n} \mathbb{I}(S_i^j \leq E(X_{n+1}, y)) > \alpha m(n+1) - m. \tag{29}$$

If $\alpha m(n+1)$ is an integer then $E(X_{n+1}, y)$ needs to be larger or equal than the $\alpha m(n+1) - m + 1$ smallest values of $(S_i^j)_{i \in [n], j \in [m]}$. If $\alpha m(n+1)$ is not an integer then $\sum_{j=1}^{m} \sum_{i=1}^{n} \mathbb{I}(S_i^j \leq E(X_{n+1}, y)) \geq \lceil \alpha m(n+1) \rceil - m$, i.e. $E(X, y)$ is larger or equal than the $\lceil \alpha m(n+1) \rceil - m$ smallest values of $S_i$. However, we have $\lceil \alpha m(n+1) \rceil - 1 = \lfloor \alpha m(n+1) \rfloor$ as $\alpha m(n+1)$ is not an integer. So $\lceil \alpha m(n+1) \rceil - m = \lfloor \alpha m(n+1) \rfloor - m + 1$. So overall we need $E(X_{n+1}, y)$ larger or equal than the $\lfloor \alpha m(n+1) \rfloor - m + 1$ smallest values of $(S_i^j)_{i \in [n], j \in [m]}$, equivalently larger that the quantile of $(S_i^j)_{i \in [n], j \in [m]}$ at level $\frac{\lfloor \alpha m(n+1) \rfloor - m + 1}{mn}$, i.e. $C(X_{n+1}) = \{k \in [K] : E(X_{n+1}, k) \geq \tau\}$ for $\tau$ defined in Equation (10).

## C  Additional results for skin condition classification

Figure 14 shows a baseline experiment using standard CP with sampled labels instead of our Monte Carlo CP from Algorithm 2. Specifically, we sample $m = 1$ and $m = 10$ labels as we would for Monte Carlo CP but apply the standard quantile from Equation (3) for calibration. Specifically, we use $\lfloor \alpha(n+1) \rfloor / n$ (Equation (3)) instead of $\lfloor \alpha m(n+1) \rfloor - m + 1 / mn$ (Equation (13)) which is used for Monte Carlo CP. For $m = 1$, these coincide; for $m > 2$, the quantiles may differ but are still close for large $n$. Thus, it is not surprising that this approach also obtains (on average) target coverage $1 - \alpha$. However, in contrast to Monte Carlo CP, this approach does not provide any coverage *guarantee* for $m > 1$.

Figure 15 shows the $p$-values of standard CP (with voted labels, left) and Monte Carlo CP (with sampled labels, right) w.r.t. to the voted labels (top) and 10 labels randomly sampled from the plausibilities (per example, bottom). CP against voted labels results in the corresponding $p$-values being uniformly distributed. However, the distribution of $p$-values corresponding to sampled labels is slightly skewed towards 0 (compare to the black line). With Monte Carlo CP, we observe the opposite: the $p$-values corresponding to voted labels are not entirely uniformly distributed while those corresponding to sampled labels are. This highlights the impact of ambiguity on the corresponding $p$-values.

Figure 16 presents complementary results to Figure 9 in the main paper. Specifically, on top, we additionally consider ties among the voted labels (i.e., there is no unique $\arg\max$ in the plausibilities $\lambda$). In the calibration examples, we break these ties randomly. At test time, however, we can decide to evaluate

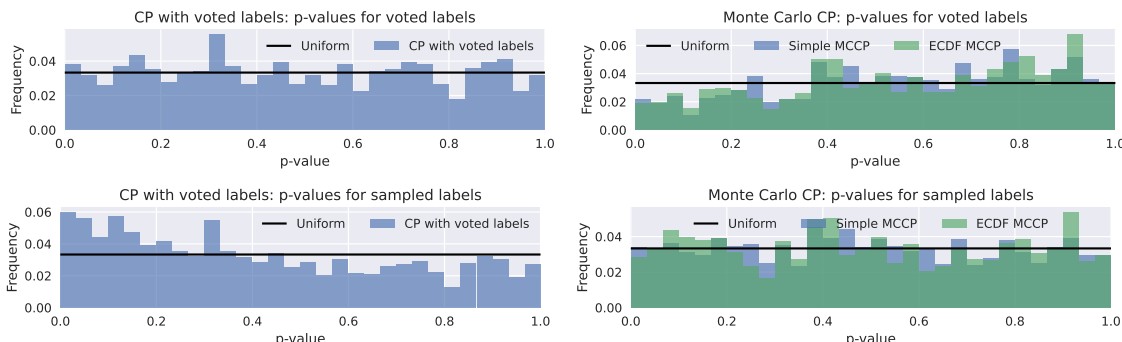

Figure 15: $p$-value histograms for CP with voted labels (left) and Monte Carlo CP (right). In both cases, we show the $p$-values for the voted labels (arg max of plausibilities, top) and labels samples from the plausibilities (bottom). Calibrating against voted labels guarantees a uniform distribution of the $p$-values shown on top, Monte Carlo calibration guarantees the same for the bottom histograms. As a result, compared to the expected uniform distribution (black), the distribution of $p$-values w.r.t. sampled labels is skewed for CP with voted labels (bottom left) while $p$-values w.r.t. to voted labels are skewed for Monte Carlo CP (top right).

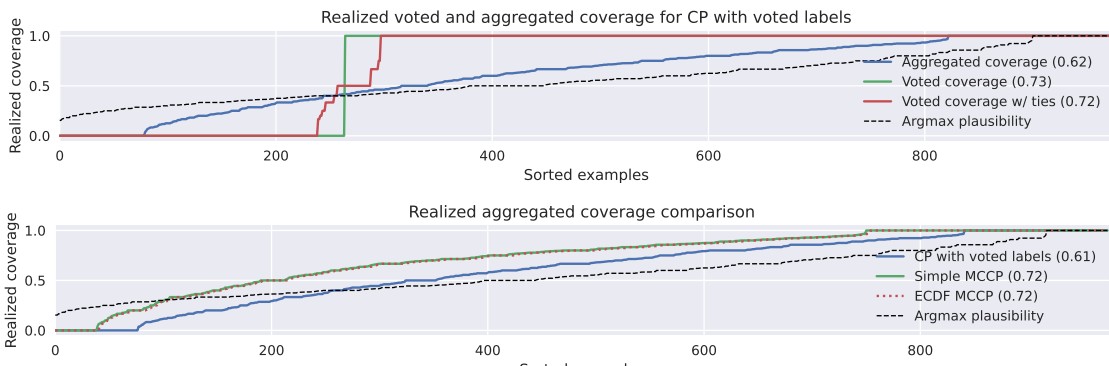

Figure 16: Top: Complementary to Figure 9, we include evaluation against voted labels while considering ties (red, $1/L_i \sum_{j=1}^{L_i} \mathbb{I}[Y_i^j \in C(X_i)]$ with $Y_i^j$ being one of $L_i$ tied labels for case $i$) which hints towards several ambiguous cases. Looking at aggregated coverage (i.e., $\sum_{y \in [K]} \lambda_{ik}\mathbb{I}[y \in C(X_i)]$) in blue, however, shows that a significant portion of examples are ambiguous and standard evaluation against the voted labels (i.e., $\mathbb{I}[\arg\max_k \lambda_{ik} \in C(X_i)]$) is unreasonable. Bottom: Aggregated coverage plot for Monte Carlo CP procedures compared to CP with voted labels, highlighting that CP with voted labels does not obtain the target aggregated coverage.

coverage proportional to the tied labels. Formally, we assume that $Y_i^1, \ldots, Y_i^{L_i}$ are the tied labels and compute $1/L_i \sum_{j=1}^{L_i} \mathbb{I}[Y_i^j \in C(X_i)]$ instead of the binary indicator $\mathbb{I}[Y_i \in C(X_i)]$ to evaluate coverage. In red, we see that quite a few examples exhibit ties and the predicted prediction sets often cover only a part of the tied labels. This is an early indicator for high ambiguity in the ground truth of this dataset. On the bottom, we additionally plot aggregated coverage for (ECDF) Monte Carlo CP in comparison to CP on voted labels. Again, CP with voted labels significantly under-estimates aggregated coverage. While both Monte Carlo approaches (with and without ECDF correction) look identical in this example, they do not have to be due to randomness in how the calibration set is further split (cf. Algorithm 2). In expectation across splits, however, we found that they coincide. This also highlights that the simple variant empirically achieves aggregated coverage $1 - \alpha$ despite only guaranteeing $1 - 2\alpha$.

Figure 17 reproduces Figure 5 on our skin condition classification case study showing the advantage of using $m > 1$ in Monte Carlo CP to reduce variation in aggregated coverage across calibration/test splits.

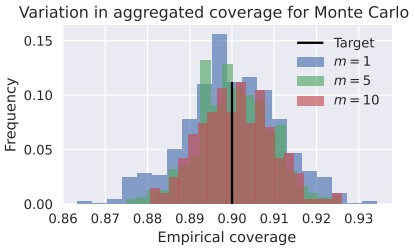

Figure 17: Empirical aggregated coverage for Monte Carlo CP with various $m$. As on our toy dataset in Figure 5, higher $m$ clearly reduces the variation in coverage across the evaluated 100 calibration/test splits despite using 50% of the 1949 examples for calibration.

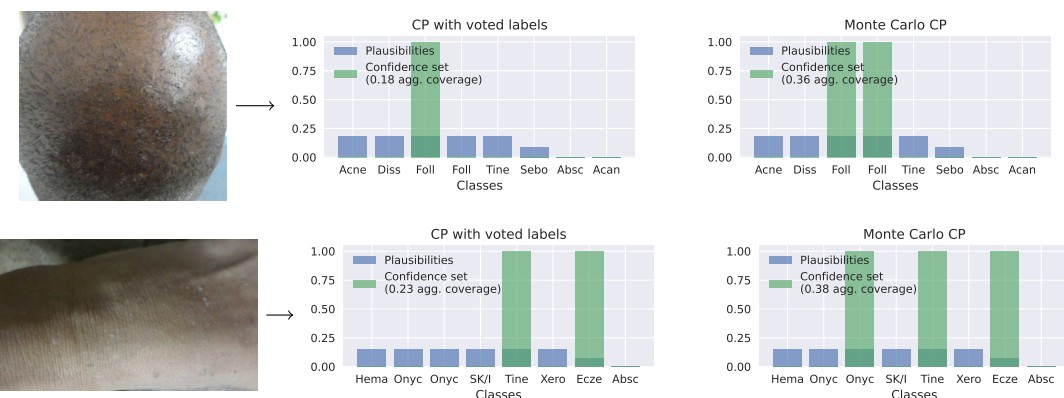

Figure 18: Additional qualitative results corresponding to Figure 11.

Figure 18 shows two additional qualitative examples where Monte Carlo CP improves results, i.e., more conditions with significant plausibility (blue) are covered (green) but important conditions are still not covered. This can be addressed using a lower confidence level such as $\alpha = 0.1$ in Figure 20.

Results of our main experiments in dermatology for $\alpha = 0.1$ can be found in Figures 19 to 20.

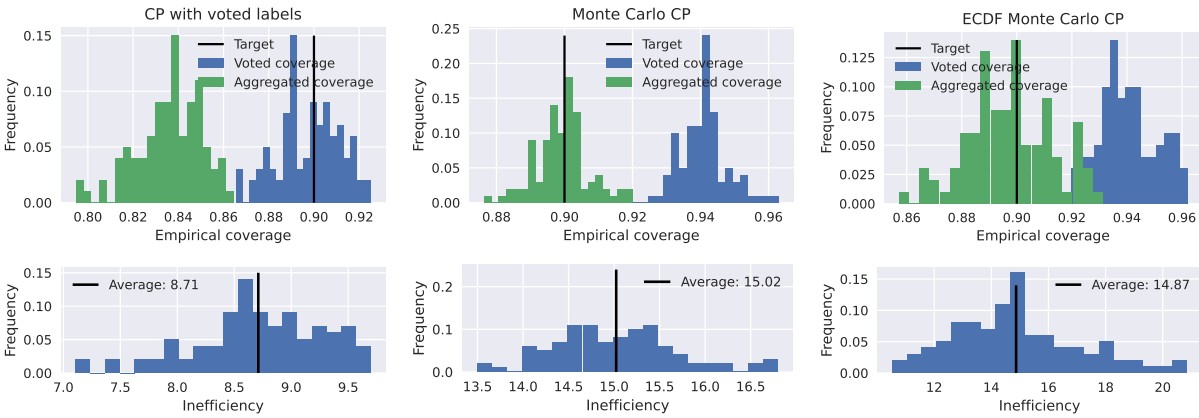

Figure 19: Results corresponding to Figure 10 with $\alpha = 0.1$.

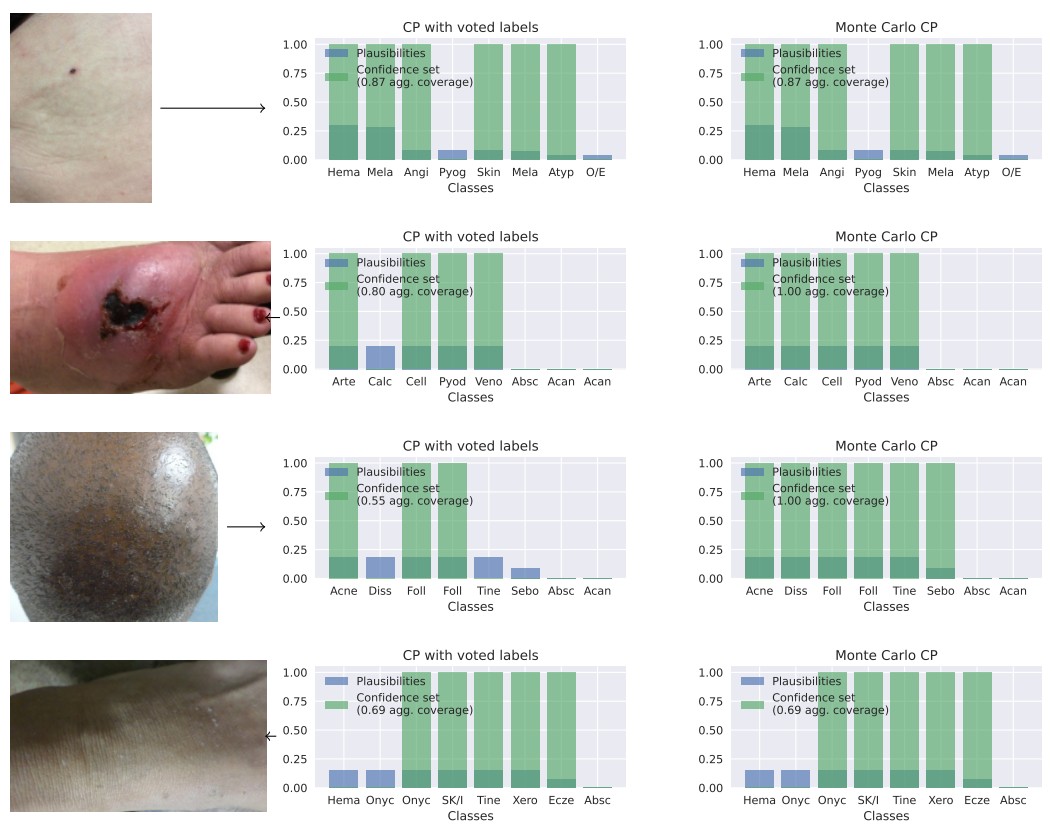

Figure 20: Additional qualitative results for $\alpha = 0.1$ corresponding to Figures 11 and 18.

