# OpenReview forum: "Conformal prediction under ambiguous ground truth"
_TMLR — Accepted by TMLR_

### Review · Reviewer_k2YP · 2023-08-04

**Summary Of Contributions:**

This paper considers conformal prediction tasks under scenarios where the ground-truth labels are inherently uncertain. By leveraging a set of label annotations from multiple experts (as an approximation of the posterior class probabilities), it first introduces a notion of plausibilities, then provides a method to estimate confidence intervals of plausibilities with associated coverage guarantee (termed as plausibility regions) by extending a standard conformal regression procedure. In addition, two strategies are proposed to reduce the estimates of plausibility regions to conformal sets of labels. Experimental results of the proposed strategies are provided with respect to both synthetic datasets of Gaussian mixtures and a real dataset for skin condition classification.

**Audience:**

Yes

**Claims And Evidence:**

Yes

**Requested Changes:**

1. Clarify the evaluation criteria for the considered problem setting.

2. Add a paragraph in the introduction to explain the contributions of your work.

3. Discuss the limitations and potential implications of deploying your methods in real-world scenarios.

4. Can you explain how the proposed methods affect the performance with respect to inputs with more deterministic labels?

**Strengths And Weaknesses:**

The paper is relatively well-written. I can easily follow the structure to understand the proposed methods. Uncertainty quantification and conformal prediction are timely research topics, particularly important for certain medical applications. This paper considers these timely topics and proposes a setting where ground-truth labels are inherently uncertain, which is new. Considering a real-world dataset of skin condition classification is also a plus.

Although the proposed setting is of practical importance, I have to say that when reading the paper, I got pretty confused about the evaluation criteria for comparisons between different conformal prediction algorithms under the new setting. It seems that the main contribution of the paper is the design of various conformal prediction algorithms which potentially work for the new setting, but it is vague what the exact improvements are when compared with baseline methods. This is a major weakness from my perspective, so I further provide my comments and questions below.

For standard conformation prediction (where the ground-truth label is deterministic), the conformal/confidence set's size and the coverage guarantee are typically considered as the evaluation criteria for comparing different methods. However, you are considering a different setting where the ground-truth labels are uncertain, so it is essential to clarify the metrics you use for the new setting. For example, if an input has uncertain labels of class 3 and class 5 with equal probabilities, could you explain what an ideal predicted conformal set for that input looks like? In addition, how to evaluate the performance of a conformal prediction algorithm with respect to an inherently uncertain data distribution? Moreover, it is unclear why we want to reduce the estimates of plausibility regions to a set of labels if the ground-truth label of the corresponding input is inherently uncertain. Please justify why this additional step is necessary for the considered setting.

Finally, I strongly recommend clarifying the contributions of your work in the introduction. The last paragraph of the introduction section just lists what you do in the paper. It is unclear what performance gain your method achieves for different datasets and what the implications are for practitioners deploying conformal prediction methods in real-world applications.

Other questions:

1. It is likely that a data distribution consists of both uncertain and deterministic components. How will your methods affect the performance of the deterministic components? Will there be a trade-off between the two components?

---

> ### Author Response · Authors · 2023-09-09
> **Reply to Reviewer k2YP**
>
> We appreciate the reviewer's feedback and have significantly revised the manuscript in response, as outlined in our general comments section.
>
> Please find below the response to your main comments.
>
> * "Clarify the evaluation criteria for the considered problem setting"
> In our new introduction, we have clarified what we are trying to achieve. Standard voting procedures ignore the label uncertainty and when using conformal prediction with such voted (i.e. top 1) labels, you become over-confident. This is illustrated in our new Figure 3 on a toy example where we have access to the true labels and calibrate with the voted labels.  In this case, the coverage w.r.t. the voted labels is at the 95% target as expected from CP but the coverage w.r.t. the true labels is at 88%.
> While we cannot provide CP procedures which guarantee coverage for the true labels (as we never observe them in our training data), we propose to better capture the label ambiguity by not using a voting procedure and better exploit the information provided by the experts.
> In our dermatology example where the labels are extremely ambiguous, calibrating w.r.t. top labels is really problematic and returns often prediction sets which ignore dangerous conditions identified by some of the experts.
> As a result, our prediction sets are generally larger than calibrating with respect to voted labels.
>
> *"Add a paragraph in the introduction to explain the contributions of your work."
> We have completely rewritten the abstract and the introduction. The rest of the paper has also been significantly updated.
>
> * "Discuss the limitations and potential implications of deploying your methods in real-world scenarios."
> We have discussed the limitations of our method in the introduction. Our method returns prediction sets which satisfies coverage guarantees for $P_{\textup{agg}} =P^X \otimes P_\{\textup{agg}}^{Y|X}$ where the "quality" of  $P_\{\textup{agg}}^{Y|X}$ is application dependent (in the dermatology example discussed here, much efforts have been done to obtain a sensible aggregation procedure, see Liu et al. 2020). If this model is poor then the prediction sets we will return will be of limited interest.  In the discussion we discuss other limitations of the methodology. The main one is that like any CP procedure, the only guarantees we have are unconditional so that we could have regions of the covariate spaces where coverage does not hold. Another limitation is that  test data might not follow the same distribution as training data. In this case, "corrected" CP methods need to be used (see Tibshirani et al., 2019) which should be adaptable to our scenario.
>
> * "Can you explain how the proposed methods affect the performance with respect to inputs with more deterministic labels?"
> In the case where the labels are quasi-deterministic, e.g.  $P_\{\textup{agg}}^{Y|X} \approx P_\{\textup{vote}^{Y|X}$, then our Monte Carlo CP performs very similarly to CP with voted labels.
>
> Other comments
>
> * " Moreover, it is unclear why we want to reduce the estimates of plausibility regions to a set of labels if the ground-truth label of the corresponding input is inherently uncertain. Please justify why this additional step is necessary for the considered setting."
>
> We have suppressed the whole section where we were using CP to obtain plausability regions to focus on the Monte Carlo CP procedures. It was not necessary to reduce them to a set of labels but it was to provide to the user a more easily interpretable prediction set.
>
> * "For example, if an input has uncertain labels of class 3 and class 5 with equal probabilities, could you explain what an ideal predicted conformal set for that input looks like?"
>
> CP provides unconditional coverage guarantees (as mentioned above) of the form $P(Y \in C(X)) \geq 1-\alpha$ where the probability is over $X,Y$ but also the calibration data (as seen as random). So for a given input X=x, we cannot say anything about $P(Y \in C(X)|X=x)$.
>
>
> We thank again the review for their comments.

---

### Review · Reviewer_ZZHz · 2023-08-16

**Summary Of Contributions:**

The paper studies the problem of conformal prediction in the setting where labels in the calibration set are not necessarily ground-truth labels, but one rather has a set of potentially disagreeing expert annotations for each datapoint. The paper provides two approaches to the problem and three algorithms (Approach 1 comes with one algorithm, and Approach 2 comes with two algorithms) that provide different guarantees. The paper compares these algorithms to standard conformal prediction in a case study of skin tone classification.

**Audience:**

Yes

**Broader Impact Concerns:**

The paper does not provide any discussion of its broader impact. While I don’t think such discussion is strictly necessary, the case study of skin condition classification somewhat cries out for at least a remark that a follow-up study should look into how the proposed method affects different demographic groups.

**Claims And Evidence:**

Yes

**Requested Changes:**

Address weaknesses W1 and W3 (critical to securing my recommendation for acceptance)

Minor fixes:
*) Figure 3 is not mentioned / discussed in the text
*) page 5, first sentence: "admits a distribution" -> admits a density
*) page 21: (Romano et al., 2019, Lemma 1) should be (Romano et al., 2019, Theorem 1)
*) page 15, first paragraph: \lambda_{ik} -> \lambda_{iy}
*) page 15, second paragraph: different -> difference

**Strengths And Weaknesses:**

Strengths:
S1) The paper studies an interesting and relevant problem
S2) it provides insightful visualizations using a running example on simple synthetic data

Weaknesses:
(I have to say that conformal prediction is not my field of expertise, still I tried to read the paper carefully)
W1) the guarantees that the algorithms provide are not w.r.t. the true distribution P, but w.r.t. the aggregation model P_{agg}. This is a major limitation in my opinion, whose consequences are not sufficiently discussed (e.g., how should one interpret the confidence sets? how does their meaningfulness depend on the quality of the plausibility model?). This limitation is also not mentioned in the abstract or introduction of the paper. There is also no discussion whether / why such limitation is necessary.
W2) The relevance of the first approach is unclear to me as it gives plausibility regions rather than confidence sets in the first place and the derived confidence sets are very inefficient in the experiments. Furthermore, the proposed approach to derive confidence sets is computationally inefficient so that it only works with three classes in the experiments.
W3) Section 3 is very hard to read in my opinion. It would benefit from
- rethinking whether the first approach shouldn’t be removed at all, or at least restructure its presentation (Sec. 3.1, 3.2, 3.3 – without any sub-subsections - are about Approach 1, Sec. 3.4, 3.4.1, 3.4.2 are about Approach 2)
- avoiding ambiguous names: the word “expected” is already overloaded, still the authors ‘refer abusively to Equation (8) as “expected” coverage’; even worse, in the following they write “expected” coverage (with quotation marks), expected coverage (without quotation marks), *expected* coverage (italic), or (expected) coverage (with parentheses) -- do these all mean the same thing?
- generally improving the presentation, e.g., formally introducing the plausibilities, discussing the conditional independence assumption of the aggregation model, etc.

---

> ### Author Response · Authors · 2023-09-09
> **Reply to Reviewer ZZHz**
>
> We appreciate the reviewer's comments and have made significant revisions to the manuscript based on their feedback, as outlined in our general comments section.
>
> We answer your main comments below.
>
> "W1) the guarantees that the algorithms provide are not w.r.t. the true distribution P, but w.r.t. the aggregation model P_{agg}. This is a major limitation in my opinion, whose consequences are not sufficiently discussed (e.g., how should one interpret the confidence sets? how does their meaningfulness depend on the quality of the plausibility model?). This limitation is also not mentioned in the abstract or introduction of the paper. There is also no discussion whether / why such limitation is necessary."
>
> We have now clarified in the abstract that the guarantees we obtain are w.r.t. $P_{\textup{agg}}$. In the introduction, we explain now in the details what this prediction set means and why we cannot expect to have prediction sets satisfying coverage guarantees w.r.t. to the true distribution. See the paragraph below we have added.
>
> "In this paper, we propose CP procedures that allow us to construct a prediction set $C(X)$ satisfying  $P_{\textup{agg}}(Y \in C(X))\geq 1-\alpha$ for $P_{\textup{agg}}=P^X  \otimes P_{\textup{agg}}^{Y|X}$ as long as one can sample from $P_{\textup{agg}}$. Note that while it would be desirable to have instead guarantees w.r.t. the distribution $P$, this is an impossible task as we never observe any data with labels sampled from $P^{Y|X}$. Whether $P_{\textup{agg}}^{Y|X}$ is a good approximation to $P^{Y|X}$ will be application dependent. Instead, the prediction set $C(X)$ outputted by our procedures is the one we could compute if the experts had access to $X$ and their opinions were aggregated through $P^{Y|X}$.
> This is the best one can hope for. We emphasize that we are not making more model assumptions than is currently made by applying CP to voted labels from $P_{\textup{vote}}^{Y|X}$. In contrast, we make the usually implicit assumptions on label collection explicit."
>
> * "W2) The relevance of the first approach is unclear to me as it gives plausibility regions rather than confidence sets in the first place and the derived confidence sets are very inefficient in the experiments. Furthermore, the proposed approach to derive confidence sets is computationally inefficient so that it only works with three classes in the experiments."
>
> We agree that the first approach we discussed in the previous version of the paper is quite intricate and not very intuitive. We have now completely suppressed this part of the paper and instead have detailed the Monte Carlo Conformal Procedures which outputs prediction sets.
>
> * "W3) Section 3 is very hard to read in my opinion."
> We have followed your advice and have significantly revised Section 3.  Section 3.1 has been significantly expanded to give some common examples of how aggregation model can be designed. We have also clarified the terminology introducing formally aggregated coverage (previously expected coverage) and voted coverage. The previous Section 3.2 and Section 3.3. which were discussing strategies to perform CP using directly the plausibilities have been suppressed as mentioned earlier and we have focused the whole manuscript on Monte Carlo conformal prediction. The new Section 3.2 (previously section 3.4) introducing Algorithm 1 has been reorganized and partially rewritten. Section 3.3 is now what used to be Section 3.4.1 and Section 3.4 introducing Algorithm 2 is what used to be section 3.4.2. They have been slightly edited. We have introduced a novel Section 3.5 and a corresponding Table 1 which summarize the theoretical and empirical properties of these algorithms. Finally, the extension of Monte Carlo CP to multi-label classification and data augmentation is now in Section 3.6 (before it was 3.5.1.) and we have shortened and clarified the techniques.
>
> * We have implemented all the other changes you suggested except the following one:
> - With respect to page 21, we kept the citation to (Romano et al., 2019, Lemma 1) instead of citing their more general Theorem 1 as this is Lemma 1 is the precise result we use.
>
> * Following your suggestion, we have now included a broader impact statement and mention your remark on a follow-up study to assess how the proposed method affects different demographic groups.
>
> We thank you again for your constructive comments.

---

### Review · Reviewer_czeu · 2023-08-26

**Summary Of Contributions:**

The standard conformal prediction builds a conformal set based on one-hot label vectors, though these one-hot vectors do not precisely provide the uncertainty on label. This paper proposes a novel approach for conformal prediction with ambiguous labels (called “plausibilities” in the paper). Assuming that the labels of a calibration set is represented in a label distribution vector, the paper introduces new terms, e.g., expected coverage in (8), expected conformity scores in (11). Based on this Algorithm 1 builds a “plausibility region” in (12) that satisfies the coverage guarantee in (13) and a reduced confidence set in (14) satisfies the coverage guarantee in (15). To improve usability and weak coverage rates, this paper further proposes Algorithm 2 and Algorithm 3. The proposed approach is evaluated on a synthetic dataset and a real dataset for skin condition classification. Along with these, the method is evaluated on two case studies.

**Audience:**

Yes

**Broader Impact Concerns:**

The Broader Impact Statement section is not presented. I think the users of this algorithm need to know the assumptions under which they can trust the output of the algorithm.

**Claims And Evidence:**

No

**Requested Changes:**

- Please provide a convincing argument that labels used in a calibration set are top-1 labels instead of sampling from a label distribution,
- Having “plausibility” labels is very expensive. Please discuss why should we use “plausibility” labels even with this high cost.
- Please discuss the relation among them and clarify the “coverage” definitions used in figures.
- Please discuss why we need “Monte Carlo conformal prediction”, which is inconsistent to the previous part of the paper and which breaks the exchangeable assumption (thus the method anyway does not provide any rigorous guarantee)

**Strengths And Weaknesses:**

This paper brings the fact that the label distribution itself is ambiguous and we need to consider the uncertainty of it in conformal prediction. This would be an interesting aspect, but I have the following major concerns on this paper’s representation and claims.

- The paper’s basic premise is that the label that we can access is actually the top-1 label (i.e., “the top-1 labels ignore the underlying ambiguity” in page 2. Based on this premise this paper claims that we have to consider a label distribution directly as a label. However, this is not true; the labels that we access for a calibration set are drawn from the label distribution, instead of the top-1 label. This means, the standard conformal prediction is already considering the ambiguity from the label uncertainty and I’m not sure if we need a novel approach.

- I agree that having “plausibility” is good, but we never get this label in practice (as it is too expensive). The “plausibility” labels obtained in skin condition classification is estimated one (from multiple experts). In this case, the estimation error will introduce another uncertainty, which is not handled by the method. So, I’m not convinced of introducing a “plausibility” label and making a new method for it.

- The paper representation along with the main claim is difficult to understand. In conformal prediction literatures, they only have a single coverage guarantee definition. However, this paper has multiple coverage guarantees, i.e., (1), (8), (13), (15) and I’m not sure which one is used in evaluation. Please discuss the relation among them and clarify the “coverage” definitions used in figures.

- “Monte Carlo conformal prediction” is introduced saying that “To avoid working with plausibility regions“ in page 8. This is inconsistent with the original claim that we need “plausibility” labels/regions. Moreover, introducing “Monte Carlo conformal prediction” breaks the exchangeability assumption, which is not desirable and undermines the usability of this approach.


Minor editorial comments:
- The caption of the first figure in Figure 1 should be “true label”?
- Prediction sets are more usual than confidence sets in conformal prediction.
- In (10), a parenthesis is not closed.
- The subscription of $\lambda$ is confusingly used. In $\lambda_i$, $i$ is the index of samples while in $\lambda_k$, $k$ is the $k$-th element of the vector $\lambda$.


I’m happy to correct my understanding if I misunderstood anything.

---

> ### Author Response · Authors · 2023-09-09
> **Reply to Reviewer czeu**
>
> We want to thank the reviewer for their comments. As explained in our general comments section, we have taken them into account to revise significantly the manuscript.
>
> * "The paper’s basic premise is that the label that we can access is actually the top-1 label (i.e., “the top-1 labels ignore the underlying ambiguity” in page 2. Based on this premise this paper claims that we have to consider a label distribution directly as a label. However, this is not true; the labels that we access for a calibration set are drawn from the label distribution, instead of the top-1 label."
>
> We respectfully disagree with you. In many applications, the labels are obtained through some voting procedures. We have now included in the new Appendix A a list of popular ML datasets which rely on such procedures to obtain the labels.  This is also illustrated in Figure 1 for CIFAR10H.
>
> * "I agree that having “plausibility” is good, but we never get this label in practice (as it is too expensive). The “plausibility” labels obtained in skin condition classification is estimated one (from multiple experts). In this case, the estimation error will introduce another uncertainty, which is not handled by the method. So, I’m not convinced of introducing a “plausibility” label and making a new method for it.
>
> As explained in the introduction and now in Section 3.1, in case where we have access to expert annotations then these expert annotations can be easily be combined to obtain plausabilities (for the dermatology example, we use the inverse rank normalization method of Liu et al. (2020)). Uncertainty about these plausabilities can also be captured by some bootstrapping mechanism (our Monte Carlo CP procedures apply directly to this setup).
>
> * "The paper representation along with the main claim is difficult to understand. In conformal prediction literatures, they only have a single coverage guarantee definition. However, this paper has multiple coverage guarantees, i.e., (1), (8), (13), (15) and I’m not sure which one is used in evaluation. Please discuss the relation among them and clarify the “coverage” definitions used in figures."
>
> We have modified and clarified the terminology in the paper. The coverage is the probability for $Y$ to be in $C(X)$ but we consider here different distributions on $(X,Y)$. We call now $P_{\textup{agg}}(Y \in C(X))$ the aggregated coverage (it was called expected coverage in the previous version) while  $P_{\textup{vote}}(Y \in C(X))$ is the voted coverage and  $P(Y \in C(X))$ would the true coverage (the true coverage cannot be estimated empirically in our settings as it assumes having access to labels distributed according to $P^{Y|X}$).
>
> * “Monte Carlo conformal prediction” is introduced saying that “To avoid working with plausibility regions“ in page 8. This is inconsistent with the original claim that we need “plausibility” labels/regions.
>
> Plausibilities are required to obtain the pseudo-labels so cannot be avoided but doing conformal prediction directly on the plausibilities is intricate as it does not provide directly prediction sets. We have suppressed this whole part of the paper in the revision to focus on Monte Carlo conformal prediction procedures.
>
> * "Moreover, introducing “Monte Carlo conformal prediction” breaks the exchangeability assumption, which is not desirable and undermines the usability of this approach. (thus the method anyway does not provide any rigorous guarantee)"
>
> We respectfully disagree with you. While the introduction of multiple pseudo-labels for each calibration data $X_i$ breaks exchangeability, the main contribution of this paper is to show that we can still provide rigorous coverage guarantees for the two proposed procedures (Algorithm 1 and Algorithm 2): this is detailed in Section 3.3 and Section 3.4 of the revision.  The properties of these methods are now also summarized in Table 1 of Section 3.5. A list of related work discussing CP techniques in scenarios where exchangeability is not satisfied can be found in Section 3.7.
>
> * "The Broader Impact Statement section is not presented. I think the users of this algorithm need to know the assumptions under which they can trust the output of the algorithm."
>
> Following your suggestion, we have now included a broader impact statement subsection in the final section of the paper.
>
> We thank you again for your comments.

---

### Author Response · Authors · 2023-09-09
**General comments**

We want to thank the reviewers for their helpful feedback. We have taken your comments into consideration and have accordingly significantly revised the paper to clarify our objectives, the properties of the developed methods as well as their limitations.

First to clarify our objectives and contributions, we have entirely rewritten the abstract and introduction. The basic premise of the paper is that in many applications the labels we have access are not from the true conditional distribution $P^{Y|X}$ but are obtained through some voting procedure based on some expert annotations, this voting procedure inducing a one-hot distribution  $P_{\textup{vote}}^{Y|X}$. We present in Appendix A a list of popular datasets which are based on such voting procedures. In simple tasks where the labels are unambiguous, we can typically reasonably assume that the voted labels correspond to the true conditional label distribution, i.e. $P_{\textup{vote}}^{Y|X}=P^{Y|X}$ .

However, in more complex problems, such as in the dermatology problem addressed in our paper, we are in an ambiguous ground truth setting and in many cases the expert opinions (given by a partial ranking) differ significantly. In such contexts, a simple voting procedure returning a one-hot distribution $P_{\textup{vote}}^{Y|X}$ ignores the inherent label uncertainty and, if conformal prediction (CP) is applied to such voted labels, it will return prediction sets which undercovers the true labels as illustrated in our toy example of Section 2.2. We argue that it is preferable to aggregate the expert opinions into a non one-hot distribution $P_{\textup{agg}}^{Y|X}$ as it is often done in practice (see e.g. Liu et al (2020)) and then apply CP in this setup. We propose procedures which allow us to construct a prediction set C(X) with rigorous coverage guarantees under the corresponding distribution $P_{\textup{agg}}=P^X \otimes P_{\textup{agg}}^{Y|X}$.  While it would be desirable to have guarantees w.r.t. the distribution $P=P^X \otimes P^{Y|X} $ instead, this is unfortunately an impossible task as we never observe any data with labels sampled from $P^{Y|X}$. (Whether $P_{\textup{agg}}^{Y|X}$ is a good approximation to $P^{Y|X}$ will be application dependent.) Instead, the prediction set $C(X)$ outputted by our procedures is the one we could compute if the experts had access to $X$ and their opinions were aggregated through $P_{\textup{agg}}^{Y|X}$.
This is the best one can hope for. We emphasize that we are not making more model assumptions than is currently made by applying CP to voted labels from $P_{vote}^{Y|X}$. In contrast, we make the usually implicit assumptions on label collection explicit. We have made very explicit our contributions in the introduction and mention again the limitations in the final Discussion section.

To better illustrate our methodology, we have presented some simple but standard models to obtain $P_{\textup{agg}}^{Y|X}$ from expert opinions in Section 3.1 and introduce more formally the plausibilities terminology. We emphasize these can be implemented very easily given expert opinions. We have then suppressed the whole Section where CP was applied to plausibilities and focused entirely on the Monte Carlo CP procedures we have proposed as this provides prediction sets which are more easily interpretable. The theoretical and empirical coverage guarantees enjoyed by the two versions of Monte Carlo CP are now summarized in Table 1 in Section 3.5.  Finally we have improved the experimental section in light of your comments. We emphasize that it is problematic using voted labels (and thus CP for voted labels) for the dermatology application we consider as many examples are very ambiguous. In this context, aggregation techniques leading to $P_{\textup{agg}}^{Y|X}$ have been proposed to capture this uncertainty by Liu et al. (2022) that we exploit in our framework.

 We have finally entirely rewritten the final Discussion section and included a Broader impact as suggested by two reviewers. A more detailed listed of the changes in available below the new abstract.

---

### Decision · Action_Editors · 2023-10-03

**Recommendation:** Accept with minor revision

**Comment:**

Following the reviewers' recommendations, the paper is worth publication in TMLR. However, one of the reviewers asked for additional comparisons between available baselines and the proposed methods:
1. Please apply the proposed method to the toy example in Figure 3 to compare with applying standard CP to voted labels
2. Please compare the proposed method to a baseline that applies standard CP to labels drawn from the given plausibilities $\lambda$ in the main experiments of the paper

**Audience:**

Yes.

**Claims And Evidence:**

The authors study conformal prediction (CP) in the context of uncertain labels due to, for example, annotator or expert disagreement. In such cases, they authors argue, CP applied to majority vote labels may result in lower-than-desired coverage of the true label. As remedy, the authors propose a method for CP applied to the (empirical) distribution of expert labels with coverage guarantees for that distribution. If this distribution is representative of the true conditional label distribution, coverage may apply also for the true label.

Reviewers appreciated that applying standard CP to voted labels may not yield the desired coverage and that the proposed method provides the claimed guarantees for the aggregated expert label distribution. However, a concern was raised about the comparison between these two methods since they have access to different data; the plausibilities $\lambda$ are only available for the proposed methods. A reviewer argued that standard CP could, in principle, be applied to the labels drawn from these plausibilities, yet this is never used as a baseline.